# SETBP1 induces transcription of a network of development genes by acting as an epigenetic hub

Rocco Piazza [1], Vera Magistroni [1], Sara Redaelli [1], Mario Mauri[1], Luca Massimino[1], Alessandro Sessa[2], Marco Peronaci [1], Maciej Lalowski[3], Rabah Soliymani[3], Caterina Mezzatesta[1], Alessandra Pirola[1], Federica Banfi[2], Alicia Rubio [2], Delphine Rea[4], Fabio Stagno[5], Emilio Usala[6], Bruno Martino[7], Leonardo Campiotti [8], Michele Merli [9], Francesco Passamonti[10], Francesco Onida[11], Alessandro Morotti [12], Francesca Pavesi[13], Marco Bregni[14], Vania Broccoli[2,15] Marc Baumann[3] & Carlo Gambacorti-Passerini[1]

SETBP1 variants occur as somatic mutations in several hematological malignancies such as atypical chronic myeloid leukemia and as de novo germline mutations in the Schinzel–Giedion syndrome. Here we show that SETBP1 binds to gDNA in AT-rich promoter regions, causing activation of gene expression through recruitment of a HCF1/KMT2A/PHF8 epigenetic complex. Deletion of two AT-hooks abrogates the binding of SETBP1 to gDNA and impairs target gene upregulation. Genes controlled by SETBP1 such as MECOM are significantly upregulated in leukemias containing SETBP1 mutations. Gene ontology analysis of deregulated SETBP1 target genes indicates that they are also key controllers of visceral organ development and brain morphogenesis. In line with these findings, in utero brain electroporation of mutated SETBP1 causes impairment of mouse neurogenesis with a profound delay in neuronal migration. In summary, this work unveils a SETBP1 function that directly affects gene transcription and clarifies the mechanism operating in myeloid malignancies and in the Schinzel–Giedion syndrome caused by SETBP1 mutations.

[1] Department of Medicine and Surgery, University of Milano-Bicocca and San Gerardo hospital, 20900 Monza, Italy. [2] Stem Cell and Neurogenesis Unit, Division of Neuroscience, San Raffaele Scientific Institute, 20132 Milan, Italy. [3] Department of Biochemistry and Developmental Biology, Faculty of Medicine, Meilahti Clinical Proteomics Core Facility, University of Helsinki, 00290 Helsinki, Finland. [4] Service d'Hématologie Adulte, Hôpital Saint-Louis, 75010 Paris, France. [5] Chair and Hematology Section, Ferrarotto Hospital, AOU Policlinico, 95123 Catania, Italy. [6] Azienda Brotzu U.O. Ematologia e CTMO, Ospedale Businco, 09121 Cagliari, Italy. [7] UO Ematologia Azienda Ospedaliera "BIANCHI MELACRINO MORELLI", 89124 Reggio Calabria, Italy. [8] Dipartimento Medicina Clinica e Sperimentale, Università Insubria, 21100 Varese, Italy. [9] Division of Hematology, University Hospital Ospedale di Circolo e Fondazione Macchi, 21100 Varese, Italy. [10] Hematology, Dipartimento di Medicina Clinica e Sperimentale, University of Varese, 21100 Varese, Italy. [11] BMT Center - Oncohematology Unit, Fondazione IRCCS Ca' Granda Ospedale Maggiore Policlinico, University of Milan, 20122 Milan, Italy. [12] Department of Clinical and Biological Sciences, University of Torino, 10043 Orbassano (Torino), Italy. [13] Hematology and Bone Marrow Transplantation Unit, IRCCS San Raffaele Scientific Institute, 20132 Milan, Italy. [14] Oncology Unit, ASST Valle Olona, Ospedale di Circolo di Busto Arsizio, 21052 Busto Arsizio, Italy. [15] CNR Institute of Neuroscience, 20129 Milan, Italy. These authors contributed equally: Rocco Piazza, Vera Magistroni, Sara Redaelli. Correspondence and requests for materials should be addressed to R.P. (email: rocco.piazza@unimib.it)

Recently, we and others demonstrated the involvement of SETBP1 mutations in several hematological malignancies[1–9]. In our work, we showed that SETBP1 somatic variants occur in a 4 amino acid mutational hotspot within the so called SKI-homology domain. This mutational hotspot is part of a degron motif that specifies substrate recognition by the cognate SCF-β-TrCP E3 ubiquitin ligase. SETBP1 mutations cause a functional loss of the degron motif targeted by SCF-β-TrCP and responsible for the short half-life of the protein. Therefore, these mutations result in an increased half-life of the mutated SETBP1 protein causing its accumulation and inhibition of the PP2A phosphatase oncosuppressor through the SETBP1–SET–PP2A axis[1].

SETBP1 mutations occurring in the same hotspot were previously found in a germline disease known as the Schinzel–Giedion syndrome (SGS)[10]. Despite the overlap between the mutations present in hematological disorders and in SGS, recent data suggest that somatic SETBP1 mutations found in leukemias are more disruptive to the degron than germline variants responsible for the onset of SGS[11]. In SGS, germline SETBP1 mutations occur as de novo variants, causing a severe phenotype characterized by mental retardation associated with distorted neuronal layering[12], multi-organ development abnormalities, and higher than normal risk of tumors[10].

Although the first studies led to a reliable characterization of SETBP1 as an oncogene, several elements of SETBP1 activity were still unclear, in particular: (1) inhibition of PP2A phosphatase alone does not explain the SETBP1-dependent phenotype of SGS[10], and (2) SETBP1 possesses three conserved AT-hooks[10], therefore suggesting a role as a DNA-binding protein.

Preliminary evidence that murine Setbp1 is able to bind to genomic DNA (gDNA) was initially given by Oakley and colleagues[13]: by transducing murine bone marrow progenitors with high titer retrovirus expressing Setbp1 followed by chromatin immunoprecipitation (ChIP) experiments, the authors demonstrated binding of Setbp1 to Hoxa9/10 promoters and upregulation of the two genes. Using a similar murine model, Vishwakarma and colleagues[14] showed binding of Setbp1 to the Runx1 promoter; this, however, was associated with down-modulation of RUNX1 expression.

In this study, we analyze the interaction between SETBP1 and gDNA on a global, unbiased scale and demonstrate that SETBP1 binds DNA in adenine-thymine (AT)-rich promoter regions, causing activation of gene expression through the recruitment of a SET1/KMT2A (MLL1) COMPASS-like complex.

## Results

**SETBP1 as a DNA-binding protein.** Cristobal et al. showed a clear role for SETBP1 as a natural inhibitor of PP2A[15]. However, SETBP1 possesses three conserved AT-hook domains[10], responsible for binding to the minor groove of AT-rich gDNA regions; thus their presence suggests a role for SETBP1 as a DNA-binding protein.

To test this hypothesis, we generated isogenic 293 FLP-In cell lines harboring wild-type (WT) and mutated (G870S) SETBP1 in fusion with the V5 tag. Other isogenic models, such as CRISPR-Cas9, could not be used given the absence of immunoprecipitation-grade anti-SETBP1 antibodies.

The G870S variant was chosen as it is the most frequent mutation found in myelodysplastic/myeloproliferative disorders, together with the D868N[1]. The G870S line showed similar SETBP1 transcript levels but increased SETBP1 protein compared to the WT one (Supplementary Figure 1a, b), as expected[1]. Anti-V5 ChIP-Seq experiments revealed a total of 3065 broad genomic regions as being bound by SETBP1-G870S (Supplementary Data 1). The presence of broad peaks occurred mainly in AT-

rich regions (Fig. 1a; mean peaks A/T content: 65.8% vs. 59% for the whole human genome; $p < 0.0001$) in cells expressing both SETBP1-G870S and SETBP1-WT. De novo motif discovery identified a SETBP1 consensus binding site (Fig. 1b; AAAATAA/T; $p = 0.002$) largely overlapping the AT-hook consensus motif of HMGA1[16] (AAAATA; http://hocomoco.autosome.ru/motif/HMGA1_HUMAN.H10MO.D), suggesting that SETBP1 binds gDNA through its AT-hook domains. Querying the Catalog of Inferred Sequence Binding Preferences (http://cisbp.ccbr.utoronto.ca/index.php)[17] for murine Setbp1 resulted in a very similar sequence, with an AAT trinucleotide as the core motif (http://cisbp.ccbr.utoronto.ca/TFreport.php?searchTF=T008692_1.02), therefore suggesting that the DNA-binding motif for SETBP1/Setbp1 is conserved across evolution. Independent ligand-fishing experiments confirmed the ability of SETBP1 to bind to gDNA (Fig. 1c).

Peak distribution analysis revealed enrichment around promoters (Fig. 1d; 58% and 65% of the total peak count reside at $+/-50$ Kb from each transcription start site (TSS) for WT and G870S, respectively). However, the binding of SETBP1 was not restricted to promoters but also present in enhancers, exonic, intronic, and intergenic regions (Fig. 1e). In line with the quantitative effect of SETBP1 mutations on SETBP1 protein stability, SETBP1-G870S and SETBP1-WT ChIP-Seq experiments revealed a global increase of SETBP1-G870S binding across all the regions tested (Fig. 1e; Jaccard $p = 0.046$).

Peak annotation identified ectopic binding of SETBP1-G870S to 277 genes (Supplementary Figure 2a-c; Supplementary Data 2) resulting in a strong functional enrichment for development-related biological processes such as nervous system, heart, and bone development (Fig. 1f). Notably, over one third of the binding regions lies within either evolutionary conserved regions, defined as 100 bp gDNA windows characterized by a human-mouse conservation ≥70%[18], or DNase I hypersensitive clusters, thus suggesting that a dysregulation in their activation could lead to significant functional consequences (Fig. 1g).

**SETBP1 upregulates target genes at the transcriptional level.** To dissect the effect of SETBP1 at the transcriptional level, we generated RNA-Seq profiles for cells overexpressing SETBP1-G870S, SETBP1-WT, as well as an Empty control. Comparative analysis of cells overexpressing G870S vs. WT revealed a very similar profile (Fig. 2a, b). This finding supports the expected quantitative mechanism of action of SETBP1 mutations, as previously proposed[1, 3, 11].

This model is further supported by the evidence that virtually all the activating SETBP1 variants identified so far fall exactly within the PEST domain and >95% within 4 amino acids of the extremely short degron linear motif and that the disruption of the PEST domain causes subsequent impairment of the SETBP1−β-TrCP axis[1, 3, 11]. To confirm this hypothesis in the context of transcriptional regulation, we analyzed the intersection between SETBP1 peaks generated in our ChIP experiments and ENCODE transcription factor binding sites (TFBSs). In the presence of mutated SETBP1, we detected an increase in the total number of shared TFBS; however, the putative binding partners did not change between SETBP1-WT and mutated lines (Supplementary Figure 3). These data highlight the presence of a quantitative rather than a qualitative effect for SETBP1 mutations, thus supporting the reduced degradation model.

Subsequent comparative RNA-Seq analysis of SETBP1-G870S vs. Empty showed the presence of 2687 differentially expressed genes (DEGs), 57% of which were downregulated and 43% upregulated (Fig. 2a, b; Supplementary Data 3). The intersection between genes bound by SETBP1 in promoter regions and DEGs

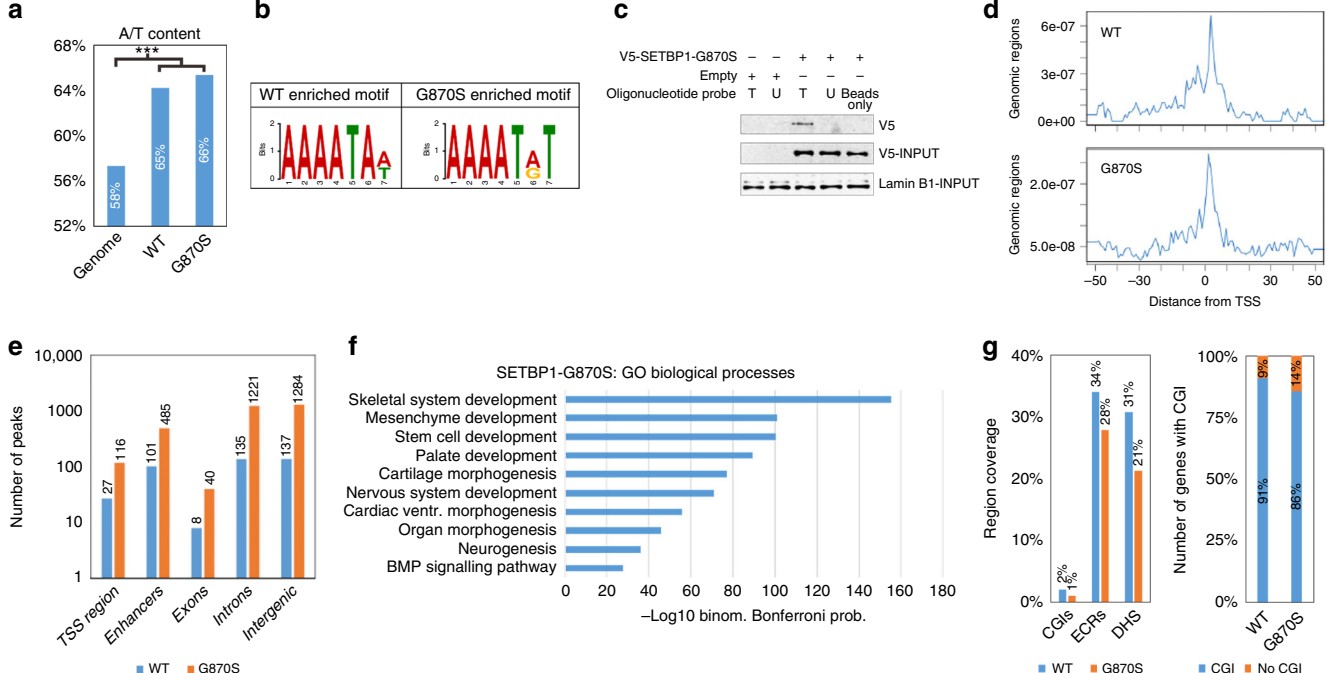

**Fig. 1** Interaction of SETBP1 with genomic DNA. **a** A/T content comparison between SETBP1 WT/G870S-binding regions and the reference genome. The frequency of A/T in SETBP1 target regions was compared with A/T frequency in the entire genome. Pearson's chi-squared test was used to test the significance of the difference between the two proportions. Actual numbers can be found in Supplementary Table 1. **b** SETBP1 consensus binding site revealed by de novo motif discovery. **c** Nuclear cell lysate oligonucleotide pulldown experiment. Target (T) and non-target (U) biotinylated probe oligonucleotides were designed according to ChIP-Seq data. Empty beads were used as control for non-specific binding. Pulldown was performed on nuclear extract from FLP-In SETBP1-G870S transfectants or Empty lines. Lamin B1 was used as a loading control. **d** Peak distribution density according to the distance from gene transcription start sites. **e** Peak quantitation in the different genomic regions, reflecting the position of binding sites relative to the next known gene. **f** Gene Ontology (GO) biological process functional enrichment of SETBP1 target genes. **g** Left, percentage of WT and G870S regions covered by CpG islands (CGIs), evolutionary conserved regions (ECRs), and DNase I hypersensitivity (DHS) at single-nucleotide resolution. Right, percentage of SETBP1 target genes having CpG islands within their promoter. ***$p < 0.001$

(false discovery rate (FDR) < 0.001) revealed 105 co-occurring genes (Fig. 2c, Supplementary Data 4). Of them, 99 (94.3%; $p < 1 \times 10^{-6}$) were upregulated, suggesting a primary role for SETBP1 as a positive inducer of gene expression (Fig. 2c). Relative quantification on a subset of target genes (*SKIDA1*, *NFE2L2*, *PDE4D*, *FBXO8*, *CEP44*, *COBLL1*, *BMP5*, *ERBB4*, *CDKN1B*) on SETBP1-WT, SETBP1-G870S, and Empty cells by mean of quantitative polymerase chain reaction (Q-PCR) confirmed the differential expression detected by RNA-Seq (Fig. 2d).

Functional annotation of DEGs and Gene Set Enrichment Analyses (Fig. 2e, f) showed significant enrichment for ontologies related to cell differentiation and tissue development, thus suggesting that SETBP1-mediated transcriptional deregulation may play an important role in the onset of SGS[10].

**SETBP1 is part of a multiprotein epigenetic complex**. AT-hook-containing proteins are often part of large chromatin remodeling complexes[19–21]. ChIP-Seq profiles of a set of histone marks associated with gene expression (H3K4me2, H3K4me3, H3K9ac, H3K27ac, and H3K36me3) revealed peak distribution enrichment around the promoter regions for all the tested marks with the exception of H3K36me3, which was enriched in gene bodies, as expected (Fig. 3a). Differential enrichment analysis in SETBP1-G870S vs. Empty showed a correlation between SETBP1 promoter occupancy and increase of H3K4me2 and H3K9ac (Fig. 3b, c; $p = 1 \times 10^{-4}$ and $p = 1.2 \times 10^{-16}$, respectively). A significant transcriptional upregulation was detected for the genes bound by SETBP1-G870S and displaying an increase in H3K4me2 and H3K9ac, corroborating the hypothesis of SETBP1

being part of a nucleosome-remodeling complex involved in transcriptional activation (Fig. 3d; Supplementary Data 5). In line with the previous findings, correlation of SETBP1-binding regions as well as related epigenetic marks between SETBP1-G870S and SETBP1-WT was nearly perfect ($r = 0.985$ and $p < 0.00001$ for anti-V5; $r = 0.873$ and $p < 0.00001$ for anti-H3K9Ac; Supplementary Figure 4).

Co-immunoprecipitation (Co-IP)/proteomics experiments directed against SETBP1 (Supplementary Data 6) revealed direct interaction of both WT and mutated SETBP1 with HCF1, a core protein of the SET1/KMT2A complex, responsible for H3K4 mono- and di-methylation[22]. These results were confirmed by independent immunoprecipitation/western blot experiments (Fig. 4a) and by acceptor photobleaching fluorescence resonance energy transfer (FRET) assays (Fig. 4b). In silico linear domain analysis[23] revealed the presence of a putative HCF1-binding motif (HBM) occurring at position 991–994 of SETBP1 (Supplementary Figure 5). Deletion of this motif in WT and mutated cells (SETBP1ΔHBM and SETBP1-G870S-ΔHBM; Supplementary Figure 6) caused a complete and specific abrogation of SETBP1/HCF1 interaction in both lines (Fig. 4b); conversely, the known SETBP1–β-TrCP[1] interaction present only in the WT protein was not affected by the deletion (Fig. 4c). To assess whether the loss of the SETBP1-HCF1 interaction could result in the impairment of the SETBP1 transcriptional machinery, we analyzed by Q-PCR the expression levels of a set of upregulated genes: the deletion of HBM resulted in a complete normalization of the expression level for all the tested genes (Fig. 4d; MECOM $p = 0.01$; BMP5 $p < 0.001$, PDE4D $p < 0.001$, ERBB4 $p = 0.036$).

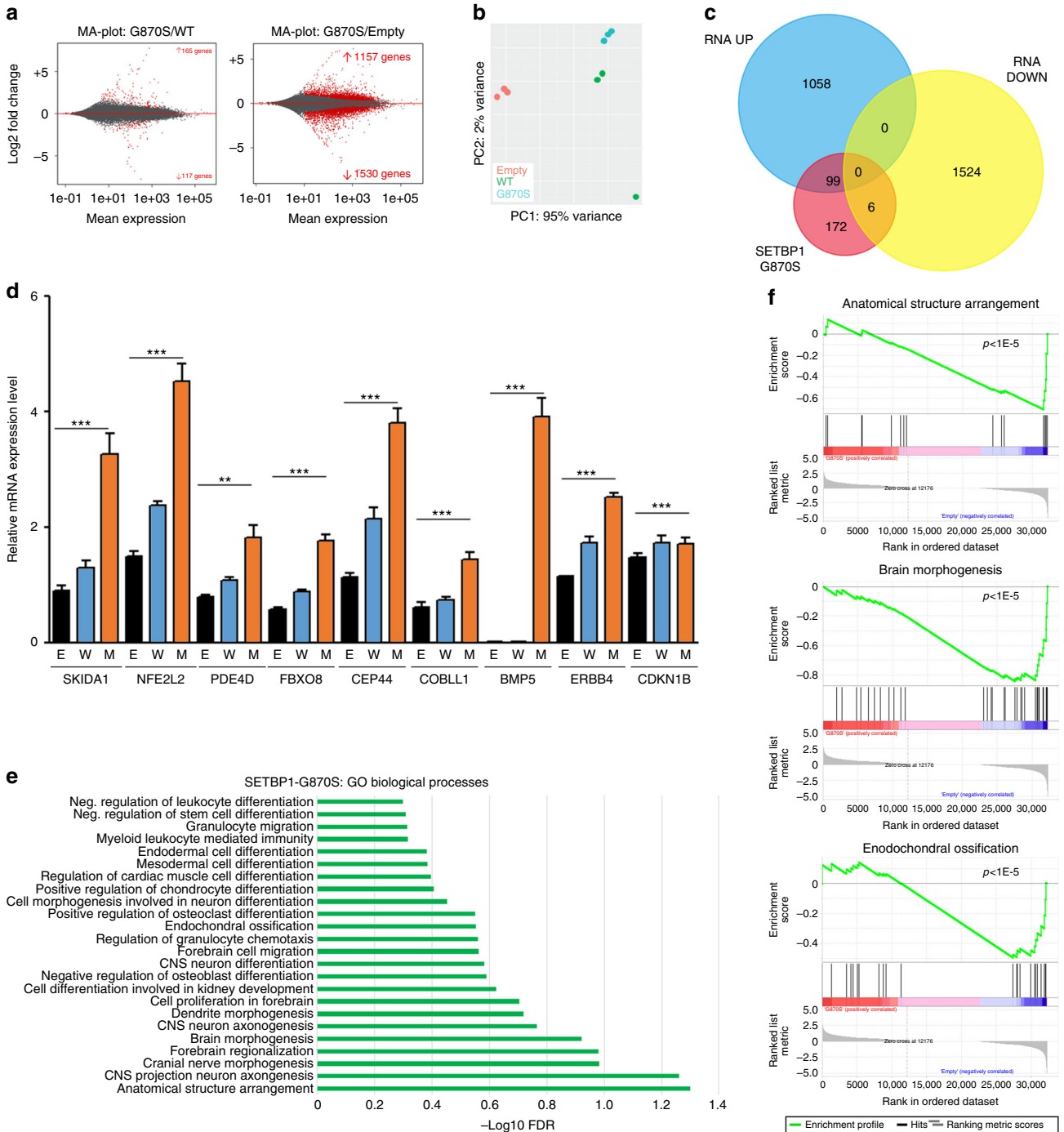

**Fig. 2** Effect of SETBP1 G870S expression at the transcriptional level. **a** MA plot showing the expression data of SETBP1-G870S vs. SETBP1-WT (left) and SETBP1-G870S vs. Empty cells (right) as a function of log ratios (M) and mean average gene counts. **b** Variance distribution of 3 Empty, 3 SETBP1-WT, and 3 SETBP1-G870S clones emphasized by Principal Component Analysis 1 (PC1) showing 95% of total variance. **c** Venn diagram showing the number of differentially expressed genes being directly bound by SETBP1 within their promoter region (red circle). **d** Q-PCR analysis of a subset of SETBP1 DEGs identified by RNA-Seq: E (Empty), W (SETBP1_wt), M (SETBP1_G870S). The housekeeping gene *GUSB* was used as an internal reference. Experiments were performed in triplicate; statistical analysis was performed using *t*-test. Error bars represent the standard error. **\*\***p < 0.01; **\*\*\***p < 0.001. **e** Dysregulated GO biological process revealed by functional enrichment analysis of the differentially expressed genes resulting from G870S mutation. **f** Gene set enrichment analysis displaying three of the most enriched categories. Genes are shown as a function of the enrichment score (*y* axis in the upper part) and relative gene expression (*x* axis)

Finally, a direct interaction between SETBP1 and KMT2A was confirmed by Co-IP experiments (Fig. 4e).

Plant homeodomain (PHD) fingers interact with methylated histone tails and are typically found in proteins responsible for the epigenetic modulation of gene expression[24]. Interestingly,

germline mutations in two PHD family members, PHF8 and PHF6, have been identified as the cause of two X-linked mental retardation syndromes, namely Siderius X-linked mental retardation[25] and Borjeson–Forssman–Lehmann syndrome[26], and somatic PHF6 mutations have been found in T cell acute

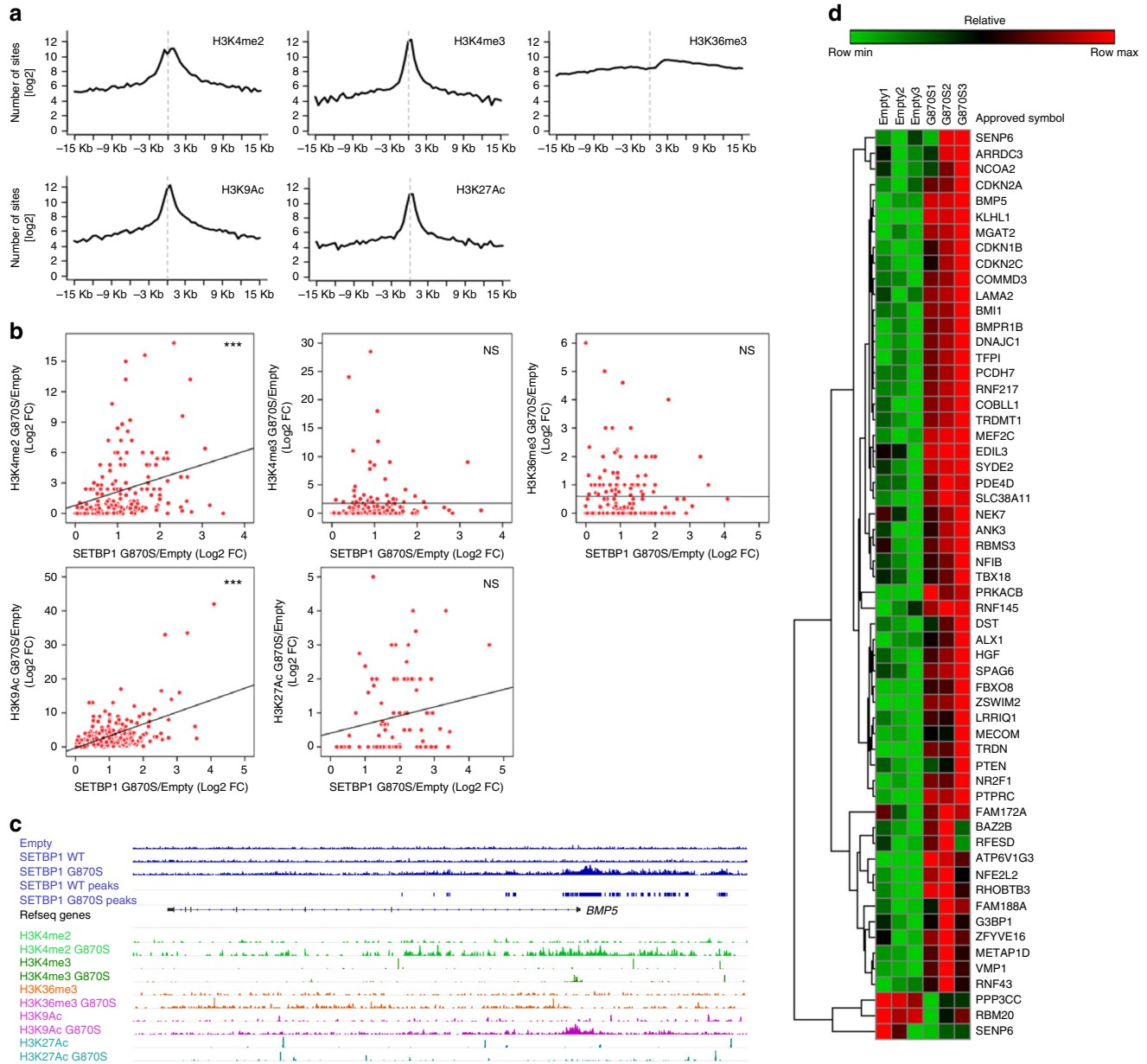

**Fig. 3** SETBP1-mediated epigenetic modulation. **a** Histone modification ChIP-Seq peak distribution densities plotted according to their distance from gene transcription start sites. **b** Epigenetic changes resulting from the presence of SETBP1-G870S expressed as function of SETBP1 differential DNA binding (ChIP-Seq G870S/Empty fold change in x axis) vs. histone modification differential enrichment (G870S/Empty fold change in y axis). ***$p < 0.001$ **c** SETBP1 ChIP-Seq coverage track and peak alignment to the hg19 reference genome are superimposed to the different histone methylation ChIP-Seq coverage tracks within the *BMP5* locus. **d** Gene expression heatmap of the subset of SETBP1 targets harboring increased H3K4me2 and H3K9ac activation marks generated on three Empty and three SETBP1-G870S clones

lymphoblastic[27] and acute myeloid leukemias (AMLs)[28]. PHF8 is also responsible for brain and craniofacial development in zebrafish[29]. While the role of PHF6 as an epigenetic modulator is less clear, PHF8 acts as a lysine demethylase. It can bind di- and tri-methylated H3K4 in the context of KMT2A complexes through its N-terminal PHD finger[24], exerting its demethylating activity on H4 Lysine 20. ChIP against H4K20me1 revealed a significant decrease in H4K20 mono-methylation in cells expressing mutated SETBP1 for all the tested SETBP1 target genes (Fig. 4f), suggesting that the SETBP1 complex possesses H4K20 demethylase activity. Co-IP experiments confirmed the interaction of SETBP1 with PHF8 and PHF6, indicating that both PHD members are part of the SETBP1 complex (Fig. 4g, h).

Taken globally, these data suggest a link between altered PHD activity and SGS phenotype.

**SETBP1 epigenetic machinery requires functional AT-hooks.** The identification of an AAAAATAA/T consensus binding motif closely resembling that of HMGA1 suggests that SETBP1 may bind to gDNA thanks to the presence of its AT-hook domains. To test this hypothesis, we generated new isogenic lines carrying deletions of the first (SETBP1-G870S$^{\Delta AT1}$), second (SETBP1-G870S$^{\Delta AT2}$), and both (SETBP1-G870S$^{\Delta AT1,2}$) AT-hook domains (Supplementary Figure 7). The third AT-hook was not deleted as it resides within the SET-binding domain. Combined disruption of AT-hooks 1 and 2 resulted in a marked reduction of mRNA

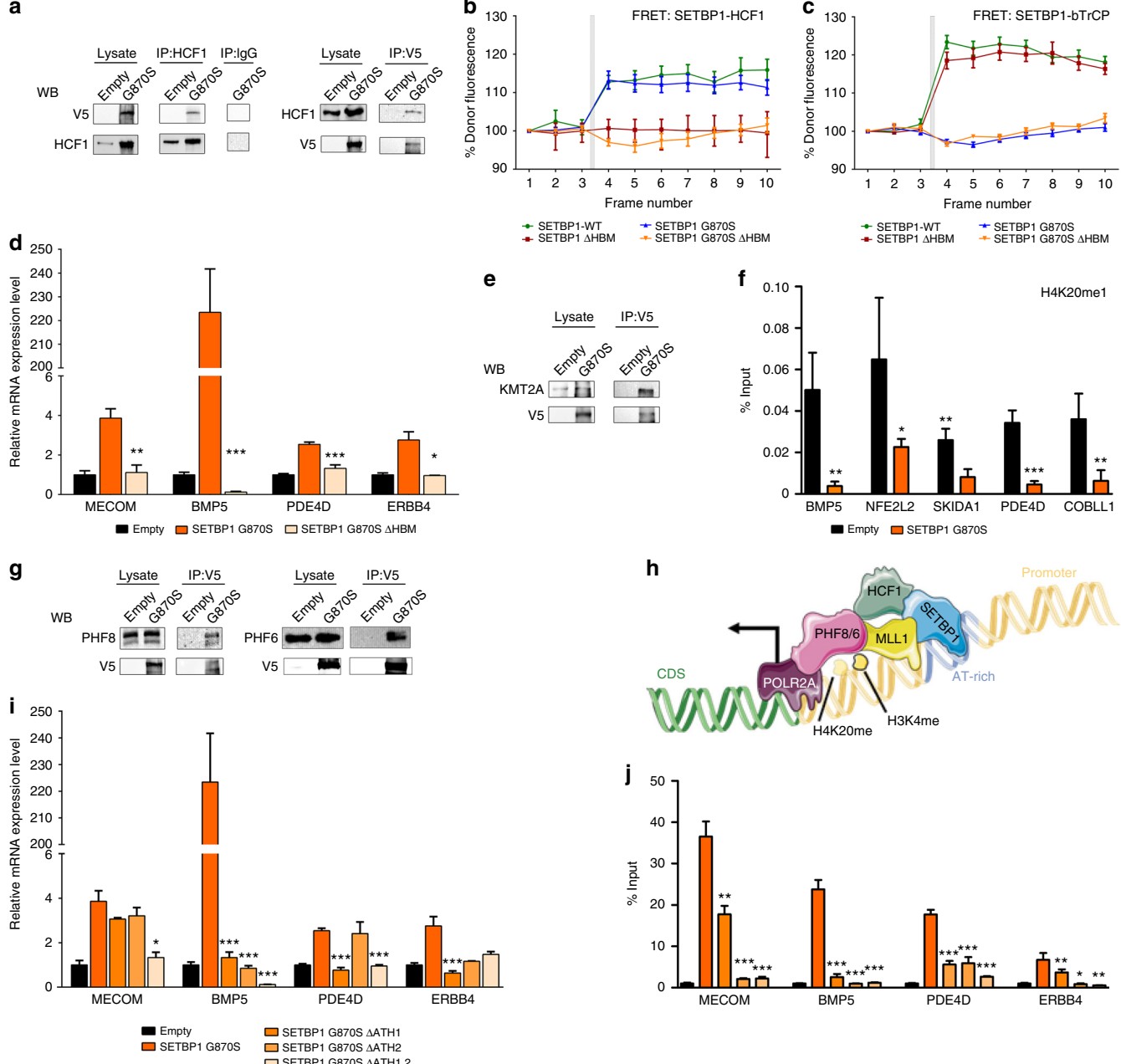

**Fig. 4** SETBP1 interacts with the SET1/KMT2A COMPASS-like complex. **a** Co-immunoprecipitation was performed against the HCF1 protein (left) or the V5 flag (right) and blotted with an anti-V5 or HCF1 antibody. **b**, **c** FRET analysis showing physical interaction between SETBP1 and HCF1. Positive FRET signal was recorded in both couples of HCF1 and SETBP1 WT or G870S (**b**, green and blue lines), conversely no FRET signal was recorded for HCF1 and SETBP1ΔHBM or G870SΔHBM (**b**, red and orange lines). FRET between β-TrCP and SETBP1 variants was assayed to demonstrate that ΔHBM did not modify the known SETBP1–β-TrCP interaction (**c**). Acceptor photobleaching was performed after the third acquired frame and indicated with gray bars in both the graphs. Bars represent the standard error of three experiments. **d** Relative expression of SETBP1 target genes as assessed by Q-PCR in empty (black), SETBP1-G870S (orange), and SETBP1-G870S-ΔHBM (light orange) lines. **e** Co-immunoprecipitation was performed against the V5 flag and blotted with an anti-KMT2A antibody. **f** ChIP against H4K20me1 followed by Q-PCR on a set of SETBP1 target genes performed on Empty (black bars) and SETBP1-G870S cells (orange bars). **g** Co-immunoprecipitation was performed against the V5 flag and blotted with anti-PHF8 and anti-PHF6 antibodies. **h** Proposed model for SETBP1 epigenetic network. **i** Relative expression of SETBP1 target genes in cells transduced with empty vector, SETBP1-G870S, or SETBP1-G870S carrying deletion of the first (ΔATH1), second (ΔATH2), and both (ΔATH1,2) AT-hooks. **j** ChIP against SETBP1-G870S in cells transduced with empty vector, SETBP1-G870S, or SETBP1-G870S carrying deletion of the first (ΔATH1), second (ΔATH2), or both (ΔATH1,2) AT-hooks, followed by Q-PCR on a set of SETBP1 target genes. In panels **d**, **f**, **i** and **j**, statistical analysis was performed using *t*-test. Bars represent the standard error of three experiments. *$p < 0.05$; **$p < 0.01$; ***$p < 0.001$

expression for all the target genes under analysis (Fig. 4i; MECOM $p = 0.014$; BMP5 $p < 0.001$; PDE4D $p < 0.001$; ERBB4-G870S$^{\Delta AT1}$ $p < 0.001$; ERBB4-G870S$^{\Delta AT1,2}$ $p = 0.1$). ChIP performed on target genes followed by Q-PCR confirmed the same effect (MECOM $p < 0.001$; BMP5 $p < 0.001$; PDE4D $p < 0.001$;

ERBB4 $p = 0.003$), indicating that SETBP1 binding to gDNA depends on the presence of functional AT-hooks (Fig. 4j).

The fact that SETBP1 is commonly found in DNase I hypersensitive cluster regions together with its ability to interact with an epigenetic activator complex containing H3K4

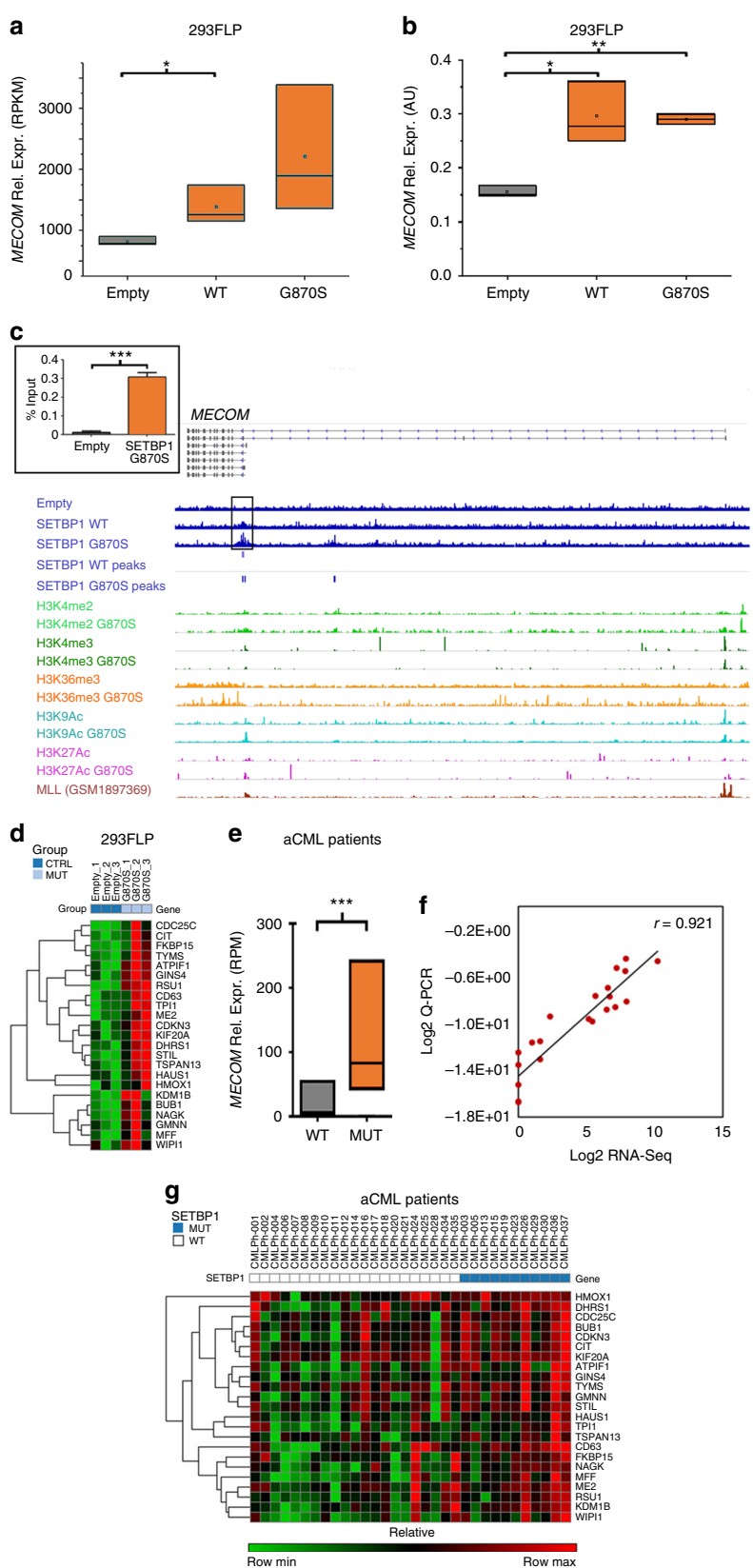

methyltransferase activity suggests that SETBP1 should be able to control gene expression through modulation of chromatin accessibility. To test this hypothesis, we generated ATAC-Seq (assay for transposase-accessible chromatin using sequencing)[30] experiments on the isogenic 293 FLP-In Empty, SETBP1-WT, and SETBP1-G870S cell lines. We assessed the ATAC-Seq signal in a region comprised between −2000 bp and +10,000 bp from the TSS of all the human genes. In line with the expected high accessibility of TSS regions, ATAC-Seq analysis revealed a peak at TSS in all the lines under test (Supplementary Figure 8a, b; Supplementary Data 7–9). Subsequently, we investigated whether a correlation could be found between the intensity of the relative ATAC-Seq signal in SETBP1-G870S vs. Empty or SETBP1-WT vs. Empty at TSS and gene bodies and the relative gene expression, as assessed by RNA-Seq in the same lines. Analysis at global gene level performed throughout a set of iterations (50) characterized by progressively more stringent differential expression fold-change filtering criteria revealed a significant linear relationship between differential ATAC-Seq signal at TSS and RNA expression for both SETBP1-G870S and SETBP1-WT (Supplementary Figure 8c). Taken globally, these data indicate that genes that are significantly upregulated in the presence of either WT or mutated SETBP1 exhibit an increased chromosomal accessibility in their TSS regions as well as in gene bodies.

**MECOM is a direct transcriptional target of SETBP1.** In 2013, Makishima and colleagues reported that the presence of mutated SETBP1 was associated with upregulation of MECOM expression[3]; however, a clear mechanistic explanation of this finding was missing. *MECOM*, located on the long arm of chromosome 3, is a transcription factor able to recruit both coactivators and corepressors[31]. It is expressed in hematopoietic stem cells, playing an important role in hematopoiesis and in hematopoietic stem cell self-renewal[32]. *MECOM* is often overexpressed, as a result of chromosomal translocations, in myelodysplastic syndromes and in approximately 10% of AML cases, being a strong negative prognostic marker for therapy response and survival[33, 34].

In line with Makishima's report, differential expression RNA-Seq (Fig. 5a; $p = 0.04$ and $p = 0.08$ for SETBP1-WT and SETBP1-G870S vs. Empty, respectively) and Q-PCR analyses (Fig. 5b; $p = 0.01$ and $p = 0.001$ for SETBP1-WT and SETBP1-G870S vs. Empty, respectively) showed that *MECOM* is upregulated in 293 FLP-In cells expressing WT or mutated SETBP1. A similar upregulation was detected in the human myeloid TF-1 cell line transduced with SETBP1-G870S (Supplementary Figure 9; $p = 0.0001$). ChIP-Seq data showed that both SETBP1-WT and SETBP1-G870S bind to the *MECOM* promoter, highlighting *MECOM* as a direct target of SETBP1 transcriptional activity (Fig. 5c). *MECOM* modulates the expression of a significant number of genes involved in hematopoietic stem cell proliferation and myeloid differentiation (Supplementary Data 10). In accordance with its previously observed transcriptional activity[35], we show here that *MECOM* target genes are also differentially expressed ($p < 0.0001$, Fig. 5d) in cells expressing SETBP1-G870S. To confirm these results in fresh leukemic cells, RNA-Seq and Q-PCR analyses were performed in 32 atypical chronic myeloid leukemia (aCML) cases (11 positive and 21 negative for *SETBP1* somatic mutations). The results show that SETBP1-positive patients express higher levels of *MECOM* (Fig. 5e, f; $p = 0.0002$) and MECOM target genes (Fig. 5g; $p < 0.0001$), with excellent correlation between RNA-Seq and Q-PCR data (Fig. 5f).

**Mutated SETBP1 delays neuronal migration to the cortex.** Upon their general developmental delay, Schinzel–Giedion patients present heavy neurological impairment characterized by microcephaly, altered neuronal layering, underdeveloped corpus callosum, ventriculomegaly, cortical atrophy, or dysplasia that cause seizures and severe mental retardation[11, 12]. Additionally, the mRNA of SETBP1 gene is expressed in both germinal (high level of expression) and differentiated area (low level) of different brain regions including cerebral cortex (Supplementary Figure 10). To gain insight into the function of the mutated form of SETBP1 in nervous system development, we transduced radial glial progenitors of cerebral cortices of E13.5 mouse embryos with expression plasmids encoding SETBP1-WT, SETBP1-G870S, SETBP1-G870S$^{\Delta AT1,2}$, and SETBP1-G870S-ΔHBM using an in utero electroporation system (Fig. 6a)[36]. The transduced cells and their progeny can be easily traced along their development thanks to the green fluorescent protein (GFP) expression. Two days after the procedure, almost the totality of the SETBP1-G870S electroporated cells remained stacked in the deep part of the developing cortical wall while many control cells (transduced with GFP only) were already migrating in the outward cortical region where post-mitotic neurons reside (Fig. 6b; Supplementary Movie 1 and 2). In addition, in SETBP1-G870S-overexpressing tissue, the apical domain of the cortical layer where the neural progenitors are located was severely disorganized as shown by aberrant localization of TBR2+ intermediate progenitors that lost their typical strip arrangement in the basal part of the ventricular zone (Fig. 6c). Five days after surgery, many control GFP+ neurons were detected in the mature cerebral mantle zone, contributing to the fiber tract of the corpus callosum (Fig. 6d, arrow), while the vast majority of cells electroporated with SETBP1-G870S were incorrectly located at the deepest cortical tissue (65 vs. 32% of the control condition; $t$-test $p < 0.001$ in bins 1–3) with only few neurons projecting to the contralateral hemisphere through the corpus callosum (Fig. 6d). In line with the proposed quantitative model for SETBP1 mutations, overexpression of SETBP1-WT showed similar, albeit reduced, effects (Fig. 6d). Overexpression of mutated SETBP1 carrying either AT-hooks 1 and 2 double

**Fig. 5** Analysis of MECOM expression and downstream targets. **a** Box plot showing the RNA-Seq differential expression analysis of *MECOM* in the 293 FLP-In Empty, SETBP1-WT, and SETBP1-G870S cell models. The top and bottom of each box represent the first and third quartile, respectively; the internal line represents the median; the dot represents the mean. Experiments were performed in triplicate. **b** Q-PCR analysis of *MECOM* expression in the 293 FLP-In Empty, SETBP1-WT, and SETBP1-G870S cell models. The top and bottom of each box represent the first and third quartile, respectively; the internal line represents the median; the small square represents the mean. Experiments were performed in triplicate; statistical analysis was performed using $t$-test. **c** SETBP1 ChIP-Seq coverage track and peak alignment to the hg19 reference genome (blue tracks) are superimposed to the different histone methylation tracks and KMT2A (MLL) ChIP-Seq coverage track[61] within the MECOM locus. The boxed histogram represents an independent ChIP experiment performed against the V5 flag in the FLP-In cells followed by a Q-PCR directed against the predicted SETBP1-G870S-binding locus on the *MECOM* promoter. ChIP was performed in triplicate; statistical analysis was performed using $t$-test. **d** Gene expression heatmap of MECOM target genes in three Empty/SETBP1-G870S FLP-In clones. **e** Differential MECOM expression as read counts per million of mapped reads (RPM) in 32 aCML patients carrying WT (21) or mutated (11) SETBP1. The top and bottom of the box represent the first and third quartile, respectively; the internal line represents the median. Statistical analysis was performed using $t$-test. **f** Linear correlation of *MECOM* expression as assessed by RNA-Seq ($x$ axis) and Q-PCR ($y$ axis). $r$ represents the Pearson linear correlation coefficient. **g** Gene expression heatmap of MECOM target genes in 32 aCML patients carrying WT (21) or mutated SETBP1 (11). Error bars represent the standard error. *$p < 0.05$; **$p < 0.01$; ***$p < 0.001$

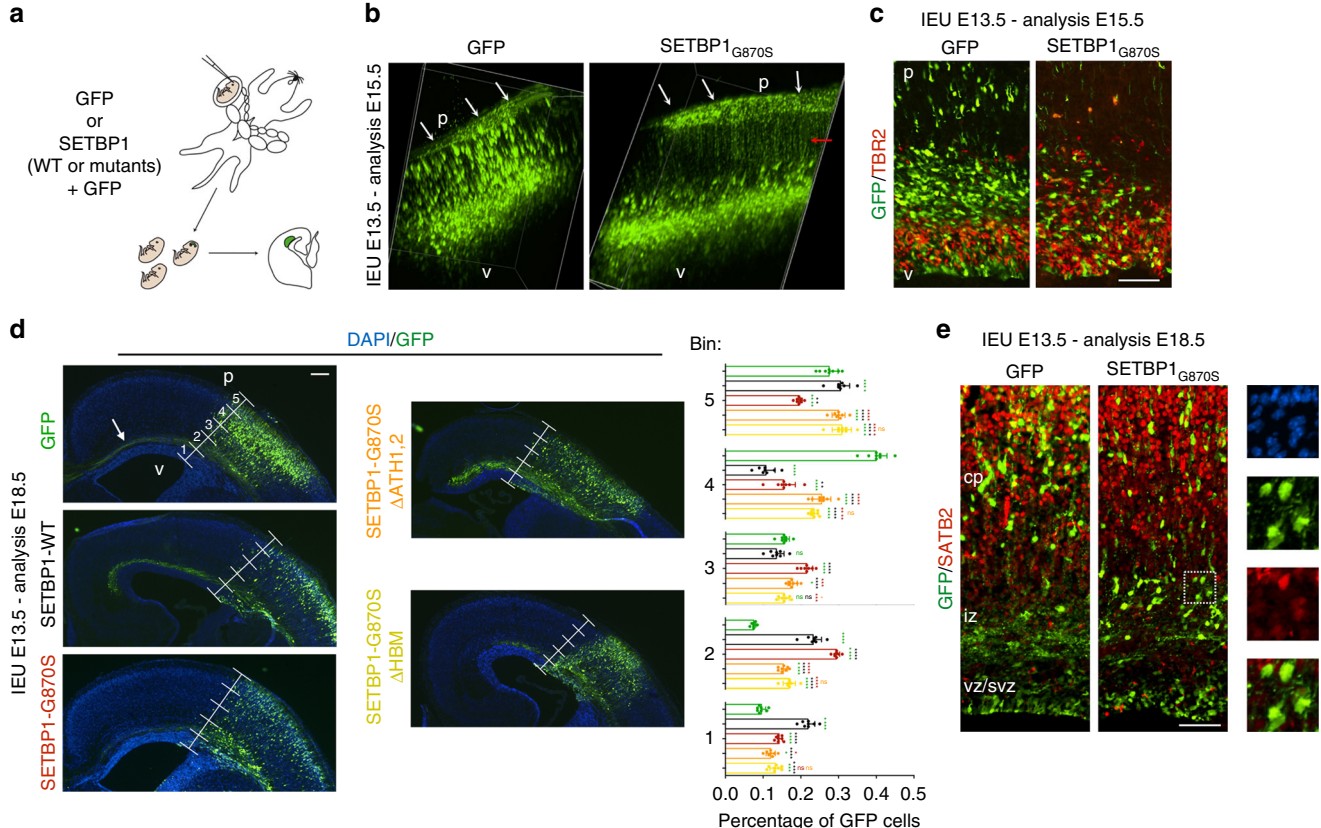

**Fig. 6** In utero electroporation of GFP and SETBP1-G870S. **a** Schematic representation of the electroporation procedure. **b** Snapshots from 3D reconstruction resulting from 2-photon microscopy on either GFP- or SETBP1-electroporated cortices (2 days) after tissue clarification (X-Clarity system) showed defects in radial migration of the SETBP1-G870S misexpressing cells. p pial side, v ventricular side. The GFP signal in the pial membrane (white arrows) and along the thickness of the SETBP1-G870S tissue (red arrow) is due to the basal processes of the GFP+ radial glia cells located in the deepest part of the organ. **c** Immunohistochemistry for GFP and TBR2 on coronal section of 2 days electroporated tissues. **d** Five-day electroporated cortices and quantification of the migration of the GFP+ cells from apical (bin #1) to pial part of the organ (bin #5); arrow indicates GFP+ corpus callosum. Statistical analysis was performed using two-way ANOVA; error bars represent standard error. *$p < 0.05$; **$p < 0.01$; ***$p < 0.001$; ***$p < 0.0001$. **e** Immunohistochemistry for GFP and SATB2 marker on coronal section of 5 days electroporated tissues. In the insets on the right, the images of SETBP1-G870S DAPI (up), GFP, SATB2, and merge of GFP/SATB2 (bottom) are shown. cp cortical plate, iz intermediate zone, svz subventricular zone, vz ventricular zone. Bars: **c**, **e**: 100 μm, **d**: 250 μm

knockout (SETBP1-G870S$^{\Delta AT1,2}$) or deletion of the HCF1-binding domain (SETBP1ΔHBM) largely restored the normal migration pattern (Fig. 6d), indicating that the presence of functional domains responsible for DNA interaction or multi-protein complex recruitment is required to modulate neuronal migration. Despite the impaired migration, the cells expressing SETBP1-G870S retained the capability of differentiating into neurons as confirmed by the expression of the mature neuronal marker SATB2 (Fig. 6e). These findings demonstrate that the increase in SETBP1 levels during brain morphogenesis has a high impact on the dynamics of both neuronal proliferation and migration that can be responsible for the neuroanatomical defects described in SGS.

## Discussion

This work describes the results of a next-generation sequencing-based, unbiased approach performed to investigate the function of SETBP1 and its pathological variant SETBP1-G870S. We demonstrate the ability of human SETBP1 to directly bind gDNA and that the binding occurs preferentially but not exclusively in gene promoter regions, as SETBP1 has also been detected in enhancers, exonic, intronic, and intergenic regions. In this work, we focused on the potential role of SETBP1 as a transcriptional

modulator, showing that SETBP1 interacts with gDNA through its AT-hook domains, forming a multiprotein complex including HCF1, KMT2A, PHF8, and PHF6, which results in increased chromatin accessibility, as assessed by ATAC-Seq (Supplementary Figure 8), and transcriptional activation. The altered transcription caused by the increased levels of SETBP1-G870S was also shown to be involved in the pathogenesis of SGS, aCML, and related myeloid malignancies, as revealed by in utero electroporation of SETBP1-G870S in ventricular central nervous system cells and by the analysis of aCML samples.

Oakley et al. previously showed that, in murine bone marrow, progenitors transduced with high-titer retrovirus-expressing Setbp1 are able to bind to Hoxa9/10 and Myb promoters and to upregulate these genes[13, 37]. Vishwakarma and colleagues[14], using a similar murine model, demonstrated binding of Setbp1 to the Runx1 promoter[14], which, however, was associated with Runx1 downmodulation. Interestingly, in our human 293 FLP-In model we found a similar downregulation (Supplementary Figure 11a). ChIP experiments, however, demonstrated only a very weak binding of SETBP1 to the human *RUNX1* promoter (Supplementary Figure 11b), likely due to the expression of SETBP1 being much lower in our model than in traditional high-titer retroviral transduction systems or to differences in species specificity. To test this hypothesis, we repeated ChIP experiments

using a transient transfection model, achieving a 4.2- and 6.9-fold increase in SETBP1-WT and G870S expression, respectively, when compared with FLP-In (Supplementary Figure 11c). Indeed, ChIP experiments performed using transient SETBP1 transfectants showed a significant increase in the binding to *RUNX1* promoter (Supplementary Figure 11d). This, however, was accompanied by an increase in *RUNX1* expression (Supplementary Figure 11e), confirming that SETBP1 promotes upregulation of gene expression and suggesting that *RUNX1* downmodulation is not a direct effect of SETBP1. Further studies will be required to assess whether the differences between our model and the murine model developed by Vishwakarma and colleagues[14] are species specific. DEG analysis between SETBP1-G870S and Empty cells in our FLP-In model and in aCML patients (11 positive and 21 negative for *SETBP1* somatic mutations) failed to reveal significant changes in the expression of *RUNX1*, *HOXA9/10*, or *MYB* (Supplementary Figure 12), corroborating the hypothesis that the effect of SETBP1 as a transcription factor is species/tissue specific.

In line with evidence indicating that KMT2A has strong H3 mono- and di-methylation but very weak tri-methylation activity[22], no significant H3K4me3 enrichment could be found in promoters occupied by SETBP1, although the intersection between SETBP1 promoter occupancy and transcriptome analysis together with the increase in H3K9Ac and the direct interaction of SETBP1 with PHF8 demethylase reveal that SETBP1 is part of a transcriptional activator complex. The exact explanation of this complex histone pattern is still unclear, given also our limited knowledge about the functions of the H3K4me2 mark. Nevertheless, several lines of evidence suggest that H3K4me2 is strongly enriched in lineage-specific gene promoters[38, 39]. The abundance of genes associated with multi-organ development among those regulated by SETBP1 is particularly interesting in the context of SGS, whose hallmark is the presence of multi-organ development abnormalities. Indeed, analysis of the Gene Ontologies associated with DEGs in SETBP1-G870S cells highlights the presence of a much stronger association with SGS than with myeloid malignancies, despite the fact that SETBP1 somatic mutations are detected in a large number of clonal myeloid disorders[1–9]. In aCML and related malignancies, the hyperactivation of the SETBP1–SET axis caused by the stabilization of mutated SETBP1 protein and leading to the inhibition of the PP2A oncosuppressor probably plays a major functional role in driving the leukemic phenotype, while the interaction with gDNA and the subsequent modulation of gene expression likely plays a critical role in the onset of SGS, in which the SETBP1-mediated inhibition of PP2A is probably insufficient to recapitulate the complex phenotype of the SGS syndrome[10]. However, the transcriptional activity of SETBP1 likely contributes to the leukemic phenotype too. In 2013, Makishima and colleagues reported that cases with SETBP1 mutations were characterized by high expression of the *MECOM* oncogene[3, 40]. Here we show that *MECOM* is a direct target of SETBP1 transcriptional activity, which explains the original observation. Interestingly, Goyama and colleagues previously showed that *SETBP1* is one of the genes whose expression is most strongly reduced in *MECOM*−/− hematopoietic cells[41], suggesting that *SETBP1* is also one of the MECOM downstream targets. Taken together, these findings suggest the presence of a positive feedback loop occurring between MECOM and SETBP1, whose biological importance and effects will require further studies. These findings highlight a potentially critical role for this transcriptional complex and its downstream effectors in the oncogenesis of SETBP1-positive myeloid disorders, therefore suggesting that a strict dichotomous model is too simplistic to fully recapitulate and explain both phenotypes. It is also important to note that, while de novo germline SETBP1 mutations

constitute the only genetic alteration in SGS, somatic SETBP1 mutations are typically present in conjunction with several other somatic events in myeloid neoplasms, in which SETBP1 mutations usually represent late events[3].

In summary, we showed here that SETBP1 is able to recruit a HCF1/KMT2A/PHF8/PHF6 transcriptional activator complex, thus identifying SETBP1 as a transcriptional activator, beyond its original SET-binding activity. Further studies will be required to gain further insights into this complex network and to dissect the functional role of SETBP1–gDNA interaction in intergenic regions.

## Methods

**Patients**. Diagnosis of aCML and related diseases was performed according to the World Health Organization 2008 classification. All patients provided written informed consent, which was approved by the institutional ethics committee. This study was conducted in accordance with the Declaration of Helsinki.

**Cell-lines**. 293T, TF-1, and the 293 FLP-In™ cell-lines were purchased from ATCC (Manassas, VA, USA), DSMZ (Braunschweig, Germany), and Thermo Fisher Scientific (Waltham, MA, USA), respectively, and maintained following the manufacturers' instructions.

**Plasmids and transfections**. Stable 293 FLP-In Empty, 293 FLP-In SETBP1wt, and 293 FLP-In SETBP1-G870S cell lines were prepared cotransfecting pOG44 (Thermo Fisher Scientific) and pFRT-SETBP1 vectors with Fugene6 reagent and were maintained in standard medium with 100 μg ml$^{-1}$ Hygromycin. Stable TF-1 cell lines were retrovirally infected using phoenix packaging cells transfected with 10 μg of MIGR1-SETBP1 Gly870Ser or WT or with empty MIGR1 vector using FuGENE6 (Promega); retroviruses were collected after 3 days of culture. Transient 293T transfectants were prepared using the pcDNA6.2-SETBP1wt, pcDNA6.2-G870S, or empty vectors[1] with Fugene6 reagent (Promega, Madison, WI, USA) following standard protocols. The deletion of the HBM site (aa991-994; NM_015559.2) within the SETBP1 coding sequence was performed with the following primers SETBP1_HCF1del_for (CAGCATTTTTCGGATTAATTTTC CGGTGCCATATATCCAGTATG) and SETBP1_HCF1del_rev (CATACTGGAT ATATGGCACCGGAAAATTAATCCGAAAAATGCTG); the deletion of the AT-hooks 1 and 2 were performed using the following primers: SETBP1_ATH1del_for (CAGTCTTACTGTGATCACTCCACTCACAGTCGAGACGATTCATG), SETBP1 ATH1del_rev (CATGAATCGTCTCGACTGTGAGTGGAGTGATCACA GTAAGACTG), SETBP1_ATH2del_for (GTAGGACTTCAGACTTGAA-GACCATGACAAAGGTGCC), and SETBP1 ATH2del_rev (GGCACCTTTGTC ATGGTCTTCAAGTCTGAAGTCCTAC) as previously described[42]. pCGN-HCF1-fl was a gift from Winship Herr (Addgene plasmid #53309).

**ChIP sequencing (ChIP-Seq)**. ChIP was performed as previously reported[43] (GEO accession number: GSE86335). Briefly, proteins were crosslinked with 0.4% formaldehyde and cells were lysed. Chromatin was fragmented with a Bioruptor sonicator system (Diagenode, SA, USA) and subsequently immunoprecipitated with H3K4me3 (ab8580, Abcam, UK), H3K4me2 (C15410035C, Diagenode), H3K36me3 (Ab9050, Abcam), H3K27Ac (Ab4729, Abcam), H3K9Ac (39137, Active Motif, Carlsbad, CA, USA), and H4K20me1 (Mab147-010, Diagenode) antibodies or anti-V5 agarose beads (Sigma-Aldrich). After immunoprecipitation, DNA was purified and libraries were prepared for sequencing following the Illumina ChIP-Seq protocol (TrueSeq ChIP library prep kit IP-202-1012) with an Illumina HiSeq2500 in single read mode (Galseq, Monza, Italy). Validation of ChIP-Seq data was performed amplifying the immunoprecipitated DNA with Sybr-Green Q-PCR. Input was used as a loading control. Primer sequences were: BMP5_Fw (CAACCCTGCTGGGAAAGAAGAG), BMP5_Rw (TCATCAAGCT AACTTAGGCACAAC), NFE2L2_Fw (AACCAGAAGAATACAATCCCAATG), NFE2L2_Rw (AAGAAGTTTCTGCTCATCCTTTGTAG), PDE4D_Fw (CCTT GAGCCAACCTTCTCCTTC), PDE4D_Rw (CACCCAAAGACATGACCAA CCTC), SKIDA1_Fw (TTCAAGTATCACGTTACTGTTTGC), SKIDA1_Rw (GTCACTTATTCAGCCACGCAGAC), COBLL1_Fw (TCTAATTGGTGGCAG GTTTAAGC), COBLL1_Rw (TGTCTGTCAGGTGTAAAGAATCATC), ERBB4_Fw (ACAAACTCCTCCAAACTGCTACTG), ERBB4_Rw (GTGATCCA TTGGAAACTGTAAATGC), RUNX1P2_Fw (CCTATGCAAACGAGCTGAGG), RUNX1P2_Rw(GCTCTATGAATGAGAGTGCCTG), MECOM_Fw (CTCCCAAATGTCTTAATCGTGTCG), and MECOM_Rw (TTCGGACCCTTTGGCTAGATTGTG).

ChIP-Seq analyses were performed using MACS v. 1.4.2[44] using the –no model parameter to skip the model building step. The ratio of the intersection and the similarity of the two genomic interval sets was calculated via bedtools Jaccard v2.26 statistics[45].

Fold enrichment of either SETBP1 or histone methylation ChIP-Seq experiments in G870S or WT background were calculated with MACS using

narrow or broad (SETBP1) peak calling parameters[46]. Transcription factor association strength and relative fold change were calculated as previously reported[47]. Downstream statistics, namely, multivariate analysis with linear correlation assumptions, were performed with the IBM SPSS statistical package.

**ATAC-Seq**. Cells (100,000/sample) were washed once in cold phosphate-buffered saline (PBS) 1×, spun at $800 \times g$ for 5' at 4 °C and resuspended in cold PBS in the presence of proteasome inhibitors and incubated on ice for 10'. Cells were centrifuged at $800 \times g$ at 4 °C for 10' and resuspended in: 2× TD buffer (25 μl), Tn5 Transposase (2.5 μl; Illumina), and water (up to 50 μl). The sample was incubated at 37 °C for 30' and purified using the SPRI AMPure XP beads. Post-tagmentation amplification was performed using Nextera primers (Illumina) and Herculase II polymerase (Agilent) using a standard protocol. Quality of the ATAC-Seq libraries was assessed using a Tape Station (Agilent) and by agar gel. Quantification was performed using a QuBit Fluorometer (ThermoFisher).

**ATAC-Seq analysis**. ATAC-Seq fastq files were initially aligned to the hg38 human reference genome using BWA[48] with standard parameters. BAM files were sorted and indexed using Samtools[49].

BAM files from individual replicates were initially merged together and subsequently processed using our ATAC-Seq tool. Briefly, all the gene start positions, gene end positions, chromosome, TSS, and gene strands were annotated (Gencode24). Coverage counts at single-nucleotide resolution were generated for each Gene/TSS from *basesBeforeTSS* (2000) to *basesAfterTSS* (10,000). Coverage at each position was then normalized using the total coverage of each input BAM. To plot ATAC-Seq heatmaps, normalized coverage data were binned (*binSize* = 200 bp) and sorted in decreasing order according to the intensity of the ATAC signal throughout the entire region (sum of the binned signals from *basesBeforeTSS* to *basesAfterTSS*). To plot line graphs, normalized counts were summed at individual bins across the entire gene set from *basesBeforeTSS* to *basesAfterTSS*. Final binned counts were then normalized by the gene set size.

To test for the presence of a correlation between chromatin accessibility and RNA expression, ATAC-Seq and RNA-Seq data generated from the same lines were compared using the following approach: normalized ATAC-Seq signal data generated for a region comprised between *basesBeforeTSS* (1500) to *basesAfterTSS* (5000) for each gene in G870S and Empty lines were initially calculated. A G870S/Empty coverage ratio was then calculated throughout the entire gene set. In parallel, normalized G870S/Empty RNA-Seq Log2 fold-change expression ratio was computed. Log2 fold-change data were filtered in *Log2_Threshold_Cycles* (50) iterations with progressively more stringent criteria, by increasing the Log2 fold-change filter from *Log2_Threshold_Start* (0) at *Log2_Threshold_Step* (+0.1 increase for each *Log2_Threshold_Cycle*) steps. For each iteration, a Pearson correlation was generated between normalized ATAC-Seq ratios and filtered RNA-Seq fold changes.

**Code availability**. Code of the ATAC-Seq tool and of the Epigenetic Marks Correlation tool will be made available upon request.

**Epigenetic marks correlation tool**. Anti-V5 and anti-H3K9Ac 293T-SETBP1-WT and G870S coverage files were used as input. Coverage was binned (BigWig) in 1000 bp regions for all the ChIP-Seq experiments. Individual bins for all the regions across the entire genome were paired and used for Pearson correlation analysis as well as for XY plots.

**Oligonucleotide pulldown assay**. Biotinylated oligonucleotides were synthesized by Metabion International AG (Steinkirchen, Germany). Target and non-target (unrelated) oligonucleotides were designed according to ChIP-Seq from SETBP1 bound (target-T, chr6: 55,742,037-55,742,136; hg19) and unbound (unrelated-U, chr6:55,775,477-55,775,576; hg19) regions. In the pulldown experiment, 25 μl of Pierce-streptavidin magnetic beads (Thermo Fisher Scientific) were washed using a magnetic stand and resuspended in 200 μl of B/W buffer (10 mM Tris HCl, pH 7.5, 1 mM EDTA, 2 M NaCl). Each double-stranded biotinylated DNA probe (200 nM) was bound to the beads for 20 min at room temperature (RT) on a rotating device. One sample without probe was used as a control for unspecific binding. Beads were washed with TE buffer and resuspended in 100 μl of BS/THES buffer (22 mM Tris HCl, pH 7.5, 4.4 mM EDTA, 8.9% Sucrose, 62 mM NaCl, 10 mM Hepes, 5 mM CaCl₂, 50 mM KCl, 12% Glycerol) supplemented with HALT protease inhibitor cocktail (Thermo Fisher Scientific). In all, 15 μg of nuclear extract from 293 FLP-In transfectants was diluted in 400 μl of BS/THES buffer and, after collecting 30 μl as INPUT fraction, was added to the beads together with 1 μg of non-specific DNA competitor LightShift Poly(dI-dC) (Thermo Scientific). Samples were incubated at 4 °C for 30 min on a rotating device. Beads were then washed four times with BS/THES buffer supplemented with 1 μg of Poly(dI-dC). Final wash was done with 500 μl of buffer (100 mM NaCl, 25 mM Tris HCl) after incubation for 1 min on a rotating device at RT. Elution of bound proteins was performed with 50 μl of Laemmli Buffer. Lamin-B1 was used as a loading control.

**Co-immunoprecipitation**. One hundred million HEK 293T cells were transiently transfected with pcDNA 6.2 V5-SETBP1-WT, SETBP1 G870S, or with the empty vector as control. Forty eight hours after transfection, cells were collected, washed twice with cold PBS, lysed in Buffer 1 (Pipes, pH 8.0, 5 mM, KCl 85 mM, NP-40 0.5%), supplemented with protease inhibitors (Halt™ Protease Inhibitor Cocktail, Thermo Fisher Scientific), kept for 10 min on ice, homogenized with douncer homogenizer (10 hits), and centrifuged at $700 \times g$ for 10 min. The pellet was then resuspended in Buffer 2 (Tris HCl, pH 8.0, 50 mM, sodium dodecyl sulfate 0.1%, deoxicholate 0.5%) supplemented with protease inhibitors, sonicated with Bioruptor Next Gen (Diagenode) (5 cycles 30 s ON, +30 s OFF) to promote gDNA disruption and centrifuged at $18,000 \times g$ for 10 min. The supernatant representing the nuclear fraction was quantified with Bradford assay and a total of 1 μg of protein was loaded on 100 μl V5-agarose beads (Sigma-Aldrich) and incubated under rotation overnight at 4 °C. Beads were then washed three times with PBS +protease inhibitors, and elution was performed with 7 M Urea, 2 M Thiourea, and 4% CHAPS for subsequent mass spectrometric analysis or with Laemmli buffer for western blot analysis. All chemicals were purchased from Sigma Aldrich.

**Immunoblot analysis**. Primary antibodies were V5 (ab27671 330 Abcam, Cambridge, UK; dilution 1:2000), HCF1 (A301-399A Bethyl Laboratories, Inc., Montgomery, TX, USA; dilution 1:1000), MLL1 (14689 Cell Signaling Technology, Danvers, MA, USA; dilution 1:1000), PHF8 (A301-772A Bethyl Laboratories, Inc., Montgomery, TX, USA; dilution 1:500), and PHF6 (A301-451A Bethyl Laboratories, Inc., Montgomery, TX, USA; dilution 1:500). Secondary antibody was anti-mouse anti-rabbit horseradish peroxidase conjugated (Biorad, Hercules, CA, USA). The uncropped scan of western blots related to PHF6 immunoprecipitation are shown in Supplementary Figure 13 as an example.

**Proteomics data analysis**. The protein samples were digested in Amicon Ultra-0.5 centrifugal filters using modified FASP method[50]. The peptides were separated with the nanoAcquity UPLC system (Waters) equipped with a 5-μm Symmetry C18 trapping column, 180 μm×20 mm, reverse-phase (Waters), followed by an analytical 1.7-μm, 75 μm×250 mm BEH-130 C18 reversed-phase column (Waters), in a single-pump trapping mode. The parameters of the HD-MSE runs were described previously[51]. Protein identifications were performed with ProteinLynx Global Server (PLGS v3.0) as described[51]. Database searches were carried out against UniProt human protein database (release_07072015, 71907 entries) with Ion Accounting algorithm and using the following parameters: peptide and fragment tolerance: automatic, maximum protein mass: 500 kDa, minimum fragment ions matches per protein: 7, minimum fragment ions matches per peptide: 3, minimum peptide matches per protein: 1, primary digest reagent: trypsin, missed cleavages allowed: 2, fixed modification: carbamidomethylation C, variable modifications: deamidation (N, Q), oxidation of Methionine (M) and FDR < 4%.

**Immunofluorescence microscopy**. Transfected HEK 293 cells, seeded on glass coverslips coated with poly-D-lysine (0.1 mg ml⁻¹), were washed twice with PBS and fixed for 10 min at 25 °C with 4% (w/v) p-formaldehyde in 0.12 M sodium phosphate buffer, pH 7.4, incubated for 1 h with primary antibodies in GDB buffer [0.02 M sodium phosphate buffer, pH 7.4, containing 0.45 M NaCl, 0.2% (w/v) bovine gelatin] followed by staining with conjugated secondary anti IgG antibodies for 1 h. After two washes with PBS, coverslips were mounted on glass slides with a 90% (v/v) glycerol/PBS solution.

**FRET**. FRET measurements were performed with the laser-induced acceptor photobleaching method[52, 53]. In our experimental set-up, FRET couples analyzed were made up of transiently transfected GFP-fused SETBP1 WT or mutated forms in combination with transiently transfected Flag-tagged β-TrCP or endogenous HCF1. Forty eight hours after transfection, HEK 293 cells were labeled with proper primary and secondary antibodies and imaged for FRET analysis. Used FRET couples were GFP signal as donor fluorochrome and Alexa Fluor 555-conjugated secondary antibodies as acceptor fluorochrome[54]. Briefly, three images were captured before bleaching in the 488 nm and the 561 nm channels using the line-by-line sequential mode without any averaging steps to reduce basal bleaching. Bleaching of the acceptor was performed within region of interest (ROI) identified in the nuclear areas with a positive colocalization between the FRET couples using 30 pulses of a full power 20 mW 561-nm laser line (each pulse 1.28 μs/pixel). After bleaching, seven images were acquired in the same channels without any time delay to obtain a full curve. The number of bleaching steps, laser intensity, and acquisition parameters were held constant throughout each experiment. FRET signal was quantified by measuring the average intensities of ROIs in the donor and acceptor fluorochrome channels before and after bleaching using the ImageJ software (http://rsbweb.nih.gov/ij/). To determine any change of fluorescence intensities not due to FRET occurring during the measurements, a distinct unbleached sentinel ROI of approximately the same size of the bleached ROI was measured in parallel, and all the results were normalized according to the background bleaching recorded in the sentinel. Proper controls were performed to verify that no artefacts were generated in the emission spectra throughout the experimental set-up due to sample overheating. Twenty measurements from three different experiments were performed for each experimental condition.

**RNA-Seq**. RNA libraries were generated starting from 1 µg of total RNA extracted from $5 \times 10^6$ cells using TRIzol (Invitrogen, Life Technologies). RNA quality was assessed by using a Tape Station instrument (Agilent). To avoid over-representation of 3′-ends, only high-quality RNA with a RNA Integrity Number ≥8 was used. RNA was processed according to the TruSeq Stranded mRNA Library Prep Kit protocol. The libraries were sequenced on an Illumina HiSeq 3000 with 76 bp paired-end reads using Illumina TruSeq technology (GEO accession number: GSE86335).

Image processing and basecall was performed using the Illumina Real Time Analysis Software. Fastq files were aligned to the human genome (GRCh38/hg39) by using STAR[55], a splice junction mapper for RNA-Seq data, together with the corresponding splice junctions Ensembl GTF annotation, using the following parameters:

--runThreadN 8 --outReadsUnmapped Fastx --outFilterType BySJout --outSAMattributes NH HI AS nM MD --outFilterMultimapNmax 20 --alignSJoverhangMin 8 --alignSJDBoverhangMin 1 --outFilterMismatchNmax 999 --outFilterMismatchNoverLmax 0.04 --alignIntronMin 20 --alignIntronMax 1000000 --alignMatesGapMax 1000000 --alignTranscriptsPerReadNmax 100000 --quantMode TranscriptomeSAM GeneCounts --limitBAMsortRAM 16620578182 --outSAMtype BAM SortedByCoordinate --chimSegmentMin 20 --chimJunctionOverhangMin 10.

**Quantitative real-time PCR**. Total RNA was extracted with Trizol reagent following standard protocol (Thermo Fisher Scientific). One microgram of RNA was used to synthesize cDNA using Reverse Transcription Reagents (Thermo Fisher Scientific) after pretreatment with DNAseI to avoid contamination from gDNA. Real-time Q-PCR was performed using TaqMan® Brilliant II QPCR Master Mix (Agilent Technologies, Santa Clara, CA, USA) on a Stratagene-MX3005P (Agilent Technologies, Santa Clara, USA) under standard conditions. The housekeeping gene *GUSB* was used as an internal reference[43]. TaqMan® Gene Expression Assays (Thermo Fisher Scientific) were used: SKIDA1 (Hs01096520), NFE2L2 (Hs00975961), PDE4D (Hs03988495), FBXO8 (Hs00942619), CEP44 (Hs00604612), COBLL1 (Hs01117513), BMP5 (Hs00234930), ERBB4 (Hs00955525), CDKN1B (Hs01597588), SETBP1 (Hs00210203), RUNX1 (Hs01021971), MECOM (Hs00602795).

**Differential expression analysis**. The gene counts generated by the --quantMode GeneCounts parameter were used to calculate gene expression. All the subsequent steps were performed using R scripts. These scripts were automatically generated through metaprogramming, using a dedicated tool (StarCounts2DESeq2) developed by RP using C# as metalanguage. Differential expression analyses were performed using DESeq2[56]. Analysis of DEGs characterized by the co-occurrence of both H3K4me2 and H3K9Ac marks was performed by multivariate regression analysis using the following filtering criteria: transcription factor and histone mark relative coverage enrichment >0.5 (log2 FC G870S/Empty); differential gene expression $p$-value < 0.05 (G870S vs. Empty, DESeq2).

**In utero electroporation**. SETBP1-G870S was cloned in the pCAG expression vector[57]. In utero electroporation was carried out at E13.5 to target the expression vectors into the ventricular radial glial cells of the mouse embryos, as previously described[58, 59]. Briefly, uterine horns of E13.5 pregnant dams were exposed by midline laparotomy after anesthetization with 312 mg kg$^{-1}$ Avertin. DNA plasmid (1 µl, corresponding to 3 µg) mixed with 0.03% fast-green dye in PBS was injected into the telencephalic vesicle using a micropipette pulled through the uterine wall and amniotic sac. Platinum tweezer-style electrodes (7 mm) were placed outside the uterus over the telencephalon, and four pulses of 40 mV for 50 ms were applied at 950 ms intervals using a BTX square wave electroporator. The uterus was then placed back into the abdomen, the cavity was filled with warm sterile PBS, and the abdominal muscle and skin incisions were closed with silk sutures.

All experiments were conducted after approval and following the guidelines of animal care and use committee from San Raffaele Scientific Institute.

**MECOM analysis**. To identify MECOM direct targets among the SETBP1 DEGs, we took advantage of previously published RNA-Seq and ChIP-Seq data in the context of MECOM overexpression[35]. The resulting genes were functionally annotated with FunRich[60] and filtered in accordance with their expression within the bone marrow (Hypergeometric enrichment test $p = 0.0013$), thus resulting in a total of 23 MECOM direct targets.

**Data availability**. All relevant sequencing data are available at the NCBI GEO Server, accession code: GSE86335.

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

## Acknowledgements

This work was supported by Associazione Italiana Ricerca sul Cancro 2013 (IG-14249) to C.G.-P., Associazione Italiana; Ricerca sul Cancro 2015 (IG-17727) to R.P., Telethon grant (GGP15096), and the Italian Ministry of Health Young investigator grant (# GR-2013-02355540) to A.S.

## Author contributions

R.P.: Conceived and designed the experiments, performed the experiments, analyzed the data, wrote the paper. V.M. and S.R.: performed experiments, analyzed the data, critically revised the manuscript. M.Mauri, M.P., and C.M.: performed experiments, critically revised the manuscript. L.M.: performed bioinformatics analyses, critically revised the manuscript. A.S., F.B., and A.R.: performed in utero electroporation experiments, critically revised the manuscript. M.L., R.S., and M.Baumann: performed mass-spectrometry experiments, critically revised the manuscript. A.P.: performed library preparation for NGS analyses, critically revised the manuscript. D.R., F.S., E.U., B.M., L.C., M.Merli, F. Passamonti, F.O., A.M., F.Pavesi, and M.Bregni: contributed reagents, critically revised the manuscript. V.B.: conceived in utero electroporation experiments, critically revised the manuscript. C.G.-P.: conceived and designed the experiments, wrote the paper.

## Additional information

**Competing interests:** The authors declare no competing interests.

