## [Peer Review File · Nature Communications]

Reviewers' comments:

Reviewer #1 (Remarks to the Author):

In this study, authors describe a novel function of SETBP1/Setbp1 protein via DNA binding as a transcriptional regulator. They demonstrated that:

- 1) SETBP1 bound to genomic DNA with a consensus AT-rich sequences,
- 2) SETBP1-bound promoters showed a significant enrichment in genes transcriptionally upregulated by forced expression of the G870S SETBP1 mutant,
- 3) Showing preferential binding to promoters with increased H2K3me2 and H2K9ac, SETBP1 is coimmunoprecipitated with HCF1 and KMT2A, depending on a putative HCF1-binding motif, which is also critical for the upregulation of SETBP1-targets,
- 4) The SETBP1-HCF1-KMRT2A complex recruited PHF8 lysine demethylase,
- 5) Two of the three AT-hooks are critical for SETBP1-binding to DNA,
- 6) In utero expression of mutant SETBP1 cause delayed neuronal migration,
- 7) MECOM is a direct target of SETBP1.

The manuscript unraveled a new aspect of SETBP1 molecular functions, showing previously unknown interactions with HCF1 and KMT2A, and other molecular mechanisms. However, this reviewer raises several concerns before it is considered for publication. Most problematic are confusing descriptions between wild-type and mutant SETBP1. To what extent, the findings are applied to both wild-type and mutant or specific to wild-type or mutant protein, and how the difference between both are responsible for the pathogenic phenotypes in SGS or SETBP1-mutated leukemias.

Major issues:

- It is frequently unclear in the main text and figures which SETBP1 is mentioned, wild-type or mutant. Please make clear whether each SETBP1 mentioned is mutant or wild-type. For example, in Fig. 1f, SETBP1 is mutant?
- The pathogenic mechanism of mutant SETBP1 is just explained by the reduced degradation and increased expression of mutant SETBP1 protein? Is there any difference between wild-type and mutant SETBP1, in terms of transcriptional regulation. Is it just the quantitative difference in transcriptional activity that discriminate wild-type and mutant SETBP1 and their effects on phenotypes? Authors mentioned the significant difference in global DNA binding between wild-type and mutant protein, but given that the substantial difference in their protein expression level, this may be due to a reduced sensitivity for wild-type protein to detect the binding sites.
- Mutant SETBP1 binding sites are shown to be enriched in DNA hypersensitivity clusters. What is the effect of SETBP1 on open chromatin regions as detected by ATAC sequencing, for example. It is affected by the mutation status of SETBP1?
- In many experiments, mutant SETBP1 was used with empty vector being used as a control, but it should be wild-type SETBP1 that is employed as a control. Is there any qualitative difference between wild-type and mutant proteins. It is not totally clear which is important, quantitative difference of the transcriptional activities mostly demonstrated for the mutant protein or qualitative one that is important for the pathogens is of leukemia or SGS. Mutation affects DNA-binding of SETBP1 protein? The current experimental design cannot answer this question.
- Statistical analysis used in Fig. 3b is questioned. These results represent real difference?
- In the in utero gene transduction experiment, the mutants lacking AT hooks, which were used in their previous studies, and the HCF1-binding site are important control, if authors claim that the abnormal transcription of the mutant is important for SGS. Also does introduction of the wild-type SETBP1 produce the same phenotype?
- Page 7, the last paragraph: because MECOM expression is largely confined to hematopoietic stem cells, comparison of MECOM expression strongly depends on the proportion of stem cells samples contains. In other words, increased MECOM expression may be not necessarily caused by increased transcription by mutant SETBP1 mutation but could be caused by an increased

proportion of stem cell fraction mutant SETBP1 might induce that may be independent from its transcriptional activity, such as PP2A suppression. So comparison of MECOM expression in SETBP1 mutated and unmutated cells should be performed on the same cell fraction (for example CD34+ cells) to control the effect of cell fraction on MECOM expression. Or is there any possibility that the mutant SETBP1 or overexpression of wild-type SETBP1 induces increased stem cell numbers?

- According to the publication of Goyama et al. (Cell Stem Cell, 2004), SETBP1 is one of the genes whose expression was most strongly reduced in MECOM^{-/-} hematopoietic cells, suggesting that SETBP1 is one of the downstream targets of MECOM, not the upstream regulator.

- What is the effect of SETBP1 on the expression of PRDM16, which is a homologue of MECOM and also deregulated in acute myeloid leukemia or MDS with t(1;3)(p36;q21), producing almost identical AML syndrome with inv(3)(q21q26) and t(3;3)(q21;q26)? PRDM16 is also implicated in successful gene therapy for CGD, together with MECOM and SETBP1.

Minor issues:

- The X axis in Fig. 3b is poorly defined. Log values of ratios in ChIP peaks between empty and mutant SETBP1?

- The results in the 3rd paragraph in Page 3 is also valid for the comparison between wild-type SETBP1 and mock? Then comparison should be made between wild-type and mutant proteins in Figs. 4d,f,I, and j.

- Enhanced MECOM expression in SETBP1 mutated cells is not totally new but have previously been reported (for example, Makishima et al, Nat. Genet 2013) and should be cited.

- Some figures have problems in readability, suffered from too small letters, no legends are provided, or inconsistency between p-values in figure and text.

Reviewer #2 (Remarks to the Author):

SETBP1 is frequently mutated in several myeloid malignancies. Most of the mutations are the missense mutations, and are accumulated to 4 amino acid residues which disrupt the binding site of SCF-beta-TrCP E3 ubiquitin ligase that targets ASXL1. In the result, these SETBP1 mutations stabilize SETBP1. It was previously reported that increased expression the mutant SETBP1 stabilizes an oncoprotein SET, leading to phosphorylation and inactivation of a tumor suppressor PP2A phosphatase. The same mutations are also found as germline mutations in Schinzel-Giedion Syndrome (SGS), characterized by several mental retardation, abnormalities of multi-organ development and increased frequencies of several tumors. In addition to its role in the signaling pathway, SETBP1 has been reported to activate transcription of HoxA9/10 via binding to the promoter regions of HoxA9/10. However, how SETBP1 is involved in transcriptional modulation has not been studied. In the present paper, the authors have identified that SETBP1 binds genome DNA through its AT-hook domains, leading to the activation of about 300 hundred genes that are related to cell differentiation and tissue development. Intriguingly, the authors have also identified that SETBP1 binds HCF1 and MLL1 as well as an H4K20 demethylase PHF8. This is an interesting paper and would attract much attention of the readers of Nature Communications. Overall, the experiments are well organized and the paper is well written. However, there are several points to be addressed.

Major comments:

1. The authors clearly demonstrated that SETBP1-G870S bound active histone marks including H3K4me2, me3, H3K9ac, and H3K27ac but not H3K36me3 (Fig. 3a). Does wildtype SETBP1 do the same?

2. The authors identified SETBP1-HCF1-MLL1 complex. They further showed that PHF8 is recruited to this complex. Is this binding inducible or stable? If this is stable association or the authors do not have any evidence that binding of PHF8 is induced by any stimulation, the word "recruit" is not appropriate to use. PHF6 is also mutated in a variety of hematological malignancies, and germline mutation of PHF6 induces developmental disorders. Did the authors also examine the binding of

PHF6 to SETBP1-HCF1-KLL1 complex?

3. There are several SETBP1 mutations and these mutations are supposed to inhibit the binding of SCF-beta-TrCP E3 ubiquitin ligase to ASXL1. The authors stated that these mutations induce functional loss of a degron region, which is confusing. In addition, they used only G807S mutant. Do the other mutations behave similarly?

Minor comments:

1. In the figure legend of Fig. 1c, the authors should clearly indicate what T or U stands for.
2. "DNA hypersensitivity" (line 112) should read "DNase I hypersensitive".

Reviewer #3 (Remarks to the Author):

In the manuscript, entitled "SETBP1 induces the transcription of a network of development genes by acting as an epigenetic hub," Piazza and colleagues provide a series of genomic and biochemical data in an attempt to show the following: (1) the chromatin associated factor SETBP1 (which is mutated in hematological malignancies and certain neurodevelopmental disorders) directly interacts with gDNA within AT rich promoters via its AT-hook domains; (2) Ectopic expression of a mutant form of SETBP1 (G870S – which is suggested to be more stable than WT SETBP1) results in altered gene expression patterns in 293 cells, some of which overlap with sites of SETBP1 (G870S) enrichment within promoters (as assessed via ChIP-seq); (3) Expression of SETBP1 (G870S) leads to altered enrichment of specific histone PTMs (e.g., H3K4me2 and H3K9ac); (4) SETBP1 (G870S) recruits the SET1/KMT2A COMPASS-like complex (e.g., HCF1, KMT2A, PHF8) to reduce H4K20me1 and promote aberrant gene expression patterns associated with cancer and/or neurodevelopmental dysfunction; (5) Overexpression of SETBP1 (G870S) leads to deficits in neuronal migration and associates with gene expression abnormalities observed in certain aCML patients). Although the authors present an interesting hypothesis and provide quite a bit of data in an attempt to bolster their claims, the paper seems to lack appropriate levels of integration, and essential controls are lacking from nearly every experiment presented. Therefore, significant revisions would be required to warrant publication in Nature Communications.

Criticisms:

- 1) Although the authors nicely begin the paper by comparing gDNA binding by WT vs. mutant SETBP1, the WT SETBP1 control, which is absolutely necessary, seems to disappear from every other set of analyses presented throughout the paper. For example, in Figure 2, the authors compare gene expression patterns between 293 cells expressing empty vector vs. SETBP1 (G870S), but very little can be interpreted about the role of this mutated protein in the absence of the WT control. Therefore, for every single analysis where SETBP1 (G870S) is being compared to empty vector, the WT control should be added.
- 2) Although one can assume from the introduction that the SETBP1 (G870S) mutant is associated with cancer and/or neurological dysfunction, nothing is mentioned as to why this specific mutation was chosen in the first place. This should be corrected.
- 3) Very few details regarding statistical parameters for sequencing studies are provided making those data challenging to assess. For example, DESeq2 is said to be used for RNA-seq differential analyses, however, it remains unclear how differential ChIP-seq data are being generated (as DESeq2 would not be entirely appropriate for use with ChIP-seq data).
- 4) Many of the conclusions drawn from sequencing studies are overstated. For example, although SETBP1 (G870S) expression (again, vs. empty) results in thousands of differentially expressed genes, only a very small % of these genes overlap with SETBP1 (G870S) enrichment sites; the authors, however, claim that the mutant is regulating gene expression through enhancement in its

promoter binding to target genes. This seems to be selective reasoning, and the authors should spend more time dissecting exactly what SETBP1 (G870S) might be doing throughout the remainder of the genome. Also, In Figure 2f, the text is almost unreadable...more emphasis on Figure preparation should be placed.

5) In Figure 3, it is very unclear as to how the authors statistically performed correlations between differential SETBP1 (G870S) enrichment vs. histone PTMs. Were Spearman rank correlations performed? If so, then the stats need to be provided.

6) The biochemical experiments in Figure 4, as presented, are inconclusive. Again, SETBP1 (G870S) was only compared to empty vector, and not WT SETBP1. Furthermore, the inputs for all IPs are not normalized, so it is impossible for the authors to claim relative enrichment.

7) For qChIP studies, it is inappropriate to display the data as "Relative Enrichment (A.U.)." These data need to be shown as % input.

8) In the absence of biophysical assessments of binding (e.g., ITC assays examining recombinant SETBP1 binding to HCF1, etc.), such claims of interactions are overstated.

9) The neuronal experiments are not interpretable in the absence of WT overexpression.

10) Figures 5 and 6 should probably be presented in reverse order (i.e., cell culture first, then in vivo analyses). Having said this, including data from both seems incohesive, and the neuronal data either need to be more fully developed or removed entirely.

In sum, although this paper has a reasonable hypothesis and provides some potentially interesting data, major revisions are needed to warrant publication at this time.

Response to Reviewers' comments:

Reviewer #1 (Remarks to the Author):

In this study, authors describe a novel function of SETBP1/Setbp1 protein via DNA binding as a transcriptional regulator. They demonstrated that:

- 1) SETBP1 bound to genomic DNA with a consensus AT-rich sequences,
- 2) SETBP1-bound promoters showed a significant enrichment in genes transcriptionally upregulated by forced expression of the G870S SETBP1 mutant,
- 3) Showing preferential binding to promoters with increased H2K3me2 and H2K9ac, SETBP1 is coimmunoprecipitated with HCF1 and KMT2A, depending on a putative HCF1-binding motif, which is also critical for the upregulation of SETBP1-targets,
- 4) The SETBP1-HCF1-KMRT2A complex recruited PHF8 lysine demethylase,
- 5) Two of the three AT-hooks are critical for SETBP1-binding to DNA,
- 6) In utero expression of mutant SETBP1 cause delayed neuronal migration,
- 7) MECOM is a direct target of SETBP1.

The manuscript unraveled a new aspect of SETBP1 molecular functions, showing previously unknown interactions with HCF1 and KMT2A, and other molecular mechanisms. However, this reviewer raises several concerns before it is considered for publication. Most problematic are confusing descriptions between wild-type and mutant SETBP1. To what extent, the findings are applied to both wild-type and mutant or specific to wild-type or mutant protein, and how the difference between both are responsible for the pathogenic phenotypes in SGS or SETBP1-mutated leukemias.

Major issues:

- It is frequently unclear in the main text and figures which SETBP1 is mentioned, wild-type or mutant. Please make clear whether each SETBP1 mentioned is mutant or wild-type. For example, in Fig. 1f, SETBP1 is mutant?

In order to avoid confusion now the specific type of SETBP1 line is indicated in Fig. 1 and in all the other figures.

- The pathogenic mechanism of mutant SETBP1 is just explained by the reduced degradation and increased expression of mutant SETBP1 protein? Is there any difference between wild-type and mutant SETBP1, in terms

of transcriptional regulation. Is it just the quantitative difference in transcriptional activity that discriminate wild-type and mutant SETBP1 and their effects on phenotypes? Authors mentioned the significant difference in global DNA binding between wild-type and mutant protein, but given that the substantial difference in their protein expression level, this may be due to a reduced sensitivity for wild-type protein to detect the binding sites.

The hypothesis of the reviewer is correct: according to our data and also to newly published results (PLoS Genet. 2017 Mar 27;13(3):e1006683), the only difference between wild-type and mutant SETBP1 is the reduced degradation. To further test this hypothesis in the context of transcriptional regulation, we analyzed the intersection between SETBP1 peaks generated in our ChIP experiments and ENCODE transcription factor binding sites (TFBSs). As shown in Suppl. Fig. 3, we detected an increase in total shared TFBS, although the putative binding partners did not change between the two conditions. In other words, we observed a quantitative rather than a qualitative increase in TFBS overlap, thus further supporting the reduced degradation model.

- Mutant SETBP1 binding sites are shown to be enriched in DNA hypersensitivity clusters. What is the effect of SETBP1 on open chromatin regions as detected by ATAC sequencing, for example. It is affected by the mutation status of SETBP1?

This is an important question that we didn't cover in the original work. To gain insight into this problem we generated ATAC-Seq data on SETBP1-WT, SETBP1-G870S and on Empty lines:

Analysis at global gene level failed to reveal any evidence of correlation (Pearson correlation < 0.1) between chromatin accessibility and gene expression, with many genes showing large differences in ATAC-Seq relative counts together with no or limited changes in gene expression. However, the same analysis performed through a set of iterations (total: 50) with progressively more stringent RNA-Seq Fold-Change filtering criteria revealed a growing linear relationship between differential ATAC-Seq signal at TSS and RNA expression for both SETBP1-G870S and SETBP1-WT (Suppl. Fig. 8c). Taken globally these data indicate that genes that are significantly upregulated in presence of either WT or mutated SETBP1 exhibit a strong increase in chromosomal accessibility in their TSS regions as well as gene bodies. This critical information is now part of the revised manuscript and we thank the reviewer for suggesting these additional experiments. These results are now described in the "SETBP1 epigenetic machinery depends on the presence of functional AT-hook domains" results paragraph.

- In many experiments, mutant SETBP1 was used with empty vector being used as a control, but it should be wild-type SETBP1 that is employed as a control. Is there any qualitative difference between wild-type and mutant proteins. It is not totally clear which is important, quantitative difference of the transcriptional activities mostly demonstrated for the mutant protein or qualitative one that is important for the pathogens is

of leukemia or SGS. Mutation affects DNA-binding of SETBP1 protein? The current experimental design cannot answer this question.

This is a very important question: the choice of the right control actually depends on the expected model. If a qualitative difference is expected, then mutant vs wild-type should be tested. However, if the difference is quantitative, all the tests performed on cells transduced/transfected with mutant vs wild-type would end-up in a dramatic underestimation of the functional effects of the variant. At present there are two lines of evidence indicating that differences between wild-type and mutated SETBP1 are strictly quantitative: the first one is data from our group and from others (PLoS Genet. 2017 Mar 27;13(3):e1006683; Nat Genet. 2013 Aug;45(8):942-6) all pointing to the disruption of the PEST domain with subsequent impairment of the SETBP1- β -TrCP axis. The second is indirect but, at least in the opinion of the authors, very strong: virtually all the somatic SETBP1 variants identified so far fall exactly within the PEST domain and >95% within 4aa of the small degron linear motif. So in order to consider the presence of a qualitative effect, we should hypothesize the degron region playing also another, yet undocumented role, which is very unlikely. This point is also supported by the fact that oncogenic somatic mutations occurring on PEST/degron domains have been found in other proteins, such as NOTCH1 (Nature. 2011 Jun 5;475(7354):101-5), but no evidence of any effect besides the stabilization of the target protein has been documented.

To strengthen this point, in the revised version we generated a comparative G870S vs WT analysis between mutated and wt under very stringent filtering criteria, to assess if a significant evidence of qualitative effects could be found. As correctly pointed-out by the reviewer this approach is however a challenging one, as any strong difference could be also potentially explained by the difference in the amount of mutated vs wt protein present in our lines (as expected by knocking out the degron). That said, the analysis of SETBP1 ChIP-seq peak overlap with the ENCODE transcription factor binding sites (Suppl. Fig. 3) and transcriptome analysis (Suppl. Fig. 4) all point toward a quantitative effect.

- Statistical analysis used in Fig. 3b is questioned. These results represent real difference?

The statistical analysis is indeed complex, however, in the opinion of the authors, a real difference among the different samples exists. The approach we used for the analysis shown in Fig. 3b is now better explained in the methods section. Briefly: fold enrichment of either SETBP1 or histone methylation ChIP-seq experiments in G870S or WT background were calculated with MACS2 using narrow or broad peak calling parameters (Feng et al., Nature Protocols, 2012). Transcription factor association strength (TFAS) and relative fold change were calculated as previously reported (Ouyang et al., PNAS, 2009). Downstream statistics, namely multivariate analysis with linear correlation assumptions, were performed with the IBM SPSS statistical package.

- In the in utero gene transduction experiment, the mutants lacking AT hooks, which were used in their previous studies, and the HCF1-binding site are important control, if authors claim that the abnormal

transcription of the mutant is important for SGS. Also does introduction of the wild-type SETBP1 produce the same phenotype?

We agree with the reviewer that, in order to support the hypothesis that abnormal transcription is important for the SGS phenotype, mutants lacking AT hooks and the HCF1-binding site are important controls for the in utero experiments, therefore we performed new experiments, which allowed us to strengthen our hypothesis: knocking-out either the AT-Hooks or the HCF1 binding site almost completely restored the correct differentiation of the cortex neurons, despite the presence of the SETBP1 G870S mutation. This highlights the role of the abnormal transcription in the onset of the SGS syndrome. A similar analysis performed using the wt SETBP1 gene confirmed again the quantitative effect of the SETBP1 mutation, with neuron progenitors overexpressing wt SETBP1 showing a phenotype that was similar, albeit decreased, to that of mutated SETBP1 (Fig. 6).

- Page 7, the last paragraph: because MECOM expression is largely confined to hematopoietic stem cells, comparison of MECOM expression strongly depends on the proportion of stem cells samples contains. In other words, increased MECOM expression may be not necessarily caused by increased transcription by mutant SETBP1 mutation but could be caused by an increased proportion of stem cell fraction mutant SETBP1 might induce that may be independent from its transcriptional activity, such as PP2A suppression. So comparison of MECOM expression in SETBP1 mutated and WT cells should be performed on the same cell fraction (for example CD34+ cells) to control the effect of cell fraction on MECOM expression. Or is there any possibility that the mutant SETBP1 or overexpression of wild-type SETBP1 induces increased stem cell numbers?

We agree that the ability to directly assess MECOM levels on CD34+ cells would strengthen this point. Unfortunately, the extremely limited availability of clinical samples prevents us from generating these data. However, FACS Analysis of aCML samples carrying mutated vs wt SETBP1 didn't show any difference in the % of CD34+ cells. Moreover, direct assessment of CD34 expression levels as well as other developmental marker on RNA-Seq data suggests that progenitor cell fraction is comparable in the two groups (See fig. below).

Heatmap and clustering analysis of the expression level (RNA-Seq) of a set of hematopoietic differentiation genes in SETBP1 WT and mutated samples.

Finally, ChIP-Seq data show a very significant enrichment of SETBP1 occupancy at MECOM promoter (Fig. 5d) coupled with upregulation of the target gene (Fig. 5a,b,c,f). Knock-out of either the AT-Hooks or the HCF1 binding site disrupts this interaction and restores normal MECOM expression levels (Fig. 4d,i,j). Taken together all these data indicate that MECOM upregulation is dependent on the interaction of mutated SETBP1 with MECOM promoter.

- According to the publication of Goyama et al. (Cell Stem Cell, 2004), SETBP1 is one of the genes whose expression was most strongly reduced in MECOM^{-/-} hematopoietic cells, suggesting that SETBP1 is one of the downstream targets of MECOM, not the upstream regulator.

The evidence shown in Goyama et al (Evi-1 Is a Critical Regulator for Hematopoietic Stem Cells and Transformed Leukemic Cells. Cell Stem Cell 2008) is extremely interesting as it suggests that a direct or indirect control of SETBP1 expression is operated by MECOM. Similarly, as discussed in the previous point, our ChIP-Seq data convincingly show that SETBP1 binds at MECOM promoter (Fig. 5d) and that this leads to MECOM upregulation (Fig. 5a,b,c,f). Taken together these findings suggest the presence of a positive feedback loop occurring between MECOM and SETBP1. This point is now discussed in the manuscript.

- What is the effect of SETBP1 on the expression of PRDM16, which is a homologue of MECOM and also deregulated in acute myeloid leukemia or MDS with t(1;3)(p36;q21), producing almost identical AML syndrome

with inv(3)(q21q26) and t(3;3)(q21;q26)? PRDM16 is also implicated in successful gene therapy for CGD, together with MECOM and SETBP1.

We agree PRDM16 could have been a good SETBP1 direct target. In addition to the well described role of PRDM16 in hematological malignancies, its germline mutation in mice was also found to result in decreased brain volumes and defective connectivity (Chuikov et al., Nature cell biology, 2003). However, no differences in PRDM16 expression were observed in patient samples harboring mutant SETBP1 (heatmap previous page; p=0.25). In accordance with these results, no SETBP1 binding within the PRDM16 promoter/enhancers was detected in HEK cells.

Minor issues:

- The X axis in Fig. 3b is poorly defined. Log values of ratios in CHIP peaks between empty and mutant SETBP1?

We agree that the X axis was poorly defined. Indeed, the X axis represents the Log2 ratio between the intensity of ChIP-Seq peaks of SETBP1-G870S vs empty. The graph is used to assess if SETBP1 promoter occupancy (x axis) correlates with a specific histone modification (y axis). Now both axes of Fig. 3b are better defined.

- The results in the 3rd paragraph in Page 3 is also valid for the comparison between wild-type SETBP1 and mock? Then comparison should be made between wild-type and mutant proteins in Figs. 4d,f,i, and j.

The comparison between mutated and WT SETBP1 is now extensively discussed in the text (e.g. the new paragraph “The effect of SETBP1 mutation G870S is quantitative” in the results section and Suppl. Fig. 3 and 4).

- Enhanced MECOM expression in SETBP1 mutated cells is not totally new but have previously been reported (for example, Makishima et al, Nat. Genet 2013) and should be cited.

We agree with the reviewer and we changed the text accordingly (Results, “MECOM is a direct transcriptional target of SETBP1” subsection and in the last paragraph of the discussion).

- Some figures have problems in readability, suffered from too small letters, no legends are provided, or inconsistency between p-values in figure and text.

Figure quality has been improved in the revised version of the manuscript.

Reviewer #2 (Remarks to the Author):

SETBP1 is frequently mutated in several myeloid malignancies. Most of the mutations are the missense mutations, and are accumulated to 4 amino acid residues which disrupt the binding site of SCF-beta-TrCP E3 ubiquitin ligase that targets ASXL1. In the result, these SETBP1 mutations stabilize SETBP1. It was previously reported that increased expression the mutant SETBP1 stabilizes an oncoprotein SET, leading to phosphorylation and inactivation of a tumor suppressor PP2A phosphatase. The same mutations are also found as germline mutations in Schinzel-Giedion Syndrome (SGS), characterized by several mental retardation, abnormalities of multi-organ development and increased frequencies of several tumors. In addition to its role in the signaling pathway, SETBP1 has been reported to activate transcription of HoxA9/10 via binding to the promoter regions of HoxA9/10. However, how SETBP1 is involved in transcriptional modulation has not been studied. In the present paper, the authors have identified that SETBP1 binds genome DNA through its AT-hook domains, leading to the activation of about 300 hundred genes that are related to cell differentiation and tissue development. Intriguingly, the authors have also identified that SETBP1 binds HCF1 and MLL1 as well as an H4K20 demethylase PHF8. This is an interesting paper and would attract much attention of the readers of Nature Communications. Overall, the experiments are well organized and the paper is well written. However, there are several points to be addressed.

Major comments:

1. The authors clearly demonstrated that SETBP1-G870S bound active histone marks including H3K4me2, me3, H3K9ac, and H3K27ac but not H3K36me3 (Fig. 3a). Does wildtype SETBP1 do the same?

This is a very important question: at present there are two lines of evidence indicating that differences between wt and mutated SETBP1 are strictly quantitative: the first one is data from our group and from others (PLoS Genet. 2017 Mar 27;13(3):e1006683; Nat Genet. 2013 Aug;45(8):942-6) all pointing to the same effect: disruption of the PEST domain and therefore impairment of the SETBP1-bTrCP axis. The second is indirect but, at least in the opinion of the authors, very strong: virtually all the somatic SETBP1 variants identified so far fall EXACTLY within the small PEST domain and >95% within 4aa of the (extremely short) degron linear motif.

In line with these considerations, if we compare the effect(s) of enforced overexpression of wt vs mutated SETBP1, results point toward a quantitative effect, with mutated SETBP1 > wt.

To clarify this point, in the revised version of our manuscript, we also generated a comparative analysis of mutated vs wt ChIP-Seq and RNA-Seq data under very stringent filtering criteria: in virtually all cases what we found points toward a quantitative effect (Results, third subsection).

2. The authors identified SETBP1-HCF1-MLL1 complex. They further showed that PHF8 is recruited to this complex. Is this binding inducible or stable? If this is stable association or the authors do not have any evidence that binding of PHF8 is induced by any stimulation, the word “recruit” is not appropriate to use. PHF6 is also mutated in a variety of hematological malignancies, and germline mutation of PHF6 induces developmental disorders. Did the authors also examine the binding of PHF6 to SETBP1-HCF1-KLL1 complex?

The hypothesis of the reviewer about a potential interaction of PHF6 with the SETBP1-HCF1-MLL1 complex is correct! To confirm it we performed new anti-PHF6 Co-IP experiments. These new experiments clearly show that also PHF6 can be part of the same complex. This important information is now part of the revised manuscript and we thank the reviewer for suggesting this additional experiment.

As indicated by the reviewer, we have no evidence of a ‘recruitment’ of PHF8 (or PHF6) to the complex, therefore we modified the text accordingly.

3. There are several SETBP1 mutations and these mutations are supposed to inhibit the binding of SCF-beta-TrCP E3 ubiquitin ligase to ASXL1. The authors stated that these mutations induce functional loss of a degron region, which is confusing. In addition, they used only G807S mutant. Do the other mutations behave similarly?

We have no specific data concerning a potential interaction of SCF-beta-TrCP or SETBP1 complex with ASXL1, as this interaction has never been detected in our work. Does the reviewer meant SETBP1 instead? If so the mechanism is the following: the degron region is a short linear motif present in the so called ‘SKI-homology domain’ of SETBP1 protein. This degron specifies substrate recognition by cognate E3 ubiquitin ligases. If the degron is functionally knocked-out by either somatic (leukemia) or germline (SGS) mutations, this results in loss of target protein (SETBP1) ubiquitination and increased target half-life. This is now better explained in the introduction (first paragraph). In this work we only used the G870S mutant, however recent data published in Plos Genetics by Rocio Acuna-Hidalgo and colleagues (PLoS Genet. 2017 Mar 27;13(3):e1006683) clearly show that all the degron-targeting SETBP1 variants work by decreasing protein degradation. This point is also discussed in detail in the text (Results, third subsection) and the recent work done by Acuna-Hidalgo and colleagues is now cited.

Minor comments:

1. In the figure legend of Fig. 1c, the authors should clearly indicate what T or U stands for.

The legend text was updated.

2. “DNA hypersensitivity” (line 112) should read “DNase I hypersensitive”.

This was corrected.

Reviewer #3 (Remarks to the Author):

In the manuscript, entitled "SETBP1 induces the transcription of a network of development genes by acting as an epigenetic hub," Piazza and colleagues provide a series of genomic and biochemical data in an attempt to show the following: (1) the chromatin associated factor SETBP1 (which is mutated in hematological malignancies and certain neurodevelopmental disorders) directly interacts with gDNA within AT rich promoters via its AT-hook domains; (2) Ectopic expression of a mutant form of SETBP1 (G870S – which is suggested to be more stable than WT SETBP1) results in altered gene expression patterns in 293 cells, some of which overlap with sites of SETBP1 (G870S) enrichment within promoters (as assessed via ChIP-seq); (3) Expression of SETBP1 (G870S) leads to altered enrichment of specific histone PTMs (e.g., H3K4me2 and H3K9ac); (4) SETBP1 (G870S) recruits the SET1/KMT2A COMPASS-like complex (e.g., HCF1, KMT2A, PHF8) to reduce H4K20me1 and promote aberrant gene expression patterns associated with cancer and/or neurodevelopmental dysfunction; (5) Overexpression of SETBP1 (G870S) leads to deficits in neuronal migration and associates with gene expression abnormalities observed in certain aCML patients). Although the authors present an interesting hypothesis and provide quite a bit of data in an attempt to bolster their claims, the paper seems to lack appropriate levels of integration, and essential controls are lacking from nearly every experiment presented. Therefore, significant revisions would be required to warrant publication in Nature Communications.

Criticisms:

1) Although the authors nicely begin the paper by comparing gDNA binding by WT vs. mutant SETBP1, the WT SETBP1 control, which is absolutely necessary, seems to disappear from every other set of analyses presented throughout the paper. For example, in Figure 2, the authors compare gene expression patterns between 293 cells expressing empty vector vs. SETBP1 (G870S), but very little can be interpreted about the role of this mutated protein in the absence of the WT control. Therefore, for every single analysis where SETBP1 (G870S) is being compared to empty vector, the WT control should be added.

This is a very important point: in the opinion of the authors the choice of the right control depends on the expected model. In presence of a qualitative difference, then Mut vs. WT should be tested and indeed this is the normal strategy used in many cell models. However, in presence of a quantitative difference, analysis of cells transduced/transfected with Mut vs WT would end-up in a dramatic underestimation of the functional

effects of the variant, as the difference between the two models at protein level can be virtually nonexistent, given the overexpression of both proteins compared to the endogenous levels. In other words, in our model we expect wt and mutant to behave very similarly, as this represents another confirmation of the quantitative model we proposed in *Nat Genet.* 2013 Jan;45(1):18-24. At present there are two lines of evidence indicating that differences between wt and mutated SETBP1 are strictly quantitative: the first one is data from our group and from others (*PLoS Genet.* 2017 Mar 27;13(3):e1006683; *Nat Genet.* 2013 Aug;45(8):942-6) all pointing to the same effect: disruption of the PEST domain and therefore impairment of the SETBP1-bTrCP axis. The second is indirect but, at least in the opinion of the authors, very strong: virtually all the somatic SETBP1 variants identified so far fall exactly within the small PEST domain and >95% within 4aa of the (extremely short) degron linear motif. So to take into account a qualitative effect, we should hypothesize another yet undocumented role for the degron motif. This is very unlikely. This point is also supported by the fact that other somatic, oncogenic mutations occurring on PEST/degron domains have been found in other proteins, such as NOTCH1 (*Nature.* 2011 Jun 5;475(7354):101-5), but no evidence of any effect besides the stabilization of the target protein (quantitative) has been documented.

To further support this point, here we generated a comparative analysis between G870S and WT under very stringent filtering criteria in order to identify the presence of significant qualitative effects: in all cases what we found pointed toward a quantitative effect, as indicated by the analysis of SETBP1 ChIP-seq peak overlap with the ENCODE transcription factor binding sites (Suppl. Fig. 3) and of transcriptome analysis (Suppl. Fig. 4)

2) Although one can assume from the introduction that the SETBP1 (G870S) mutant is associated with cancer and/or neurological dysfunction, nothing is mentioned as to why this specific mutation was chosen in the first place. This should be corrected.

In this work we only used the G870S mutant: the reason we chose the G870S variant is that it is the most frequent mutation occurring in aCML and CMML cases (*Nat Genet.* 2013 Jan;45(1):18-24). This is now indicated in the revised manuscript (Results section, second paragraph). However, recent data published in *Plos Genetics* by Rocio Acuna-Hidalgo and colleagues (*PLoS Genet.* 2017 Mar 27;13(3):e1006683) clearly show that all the degron-targeting variants are functionally identical. This is now indicated in the revised manuscript (Results section, third subsection).

3) Very few details regarding statistical parameters for sequencing studies are provided making those data challenging to assess. For example, DESeq2 is said to be used for RNA-seq differential analyses, however, it remains unclear how differential ChIP-seq data are being generated (as DESeq2 would not be entirely appropriate for use with ChIP-seq data).

We agree with the reviewer: the data provided in the first version of the manuscript about the statistical and bioinformatics methods were very limited. Indeed we never used DESeq2 for ChIP-Seq data but only for

RNA-Seq analysis (MACS was used for ChIP-Seq analyses). This is now more clearly explained in the methods section (Chromatin Immunoprecipitation sequencing subsection).

4) Many of the conclusions drawn from sequencing studies are overstated. For example, although SETBP1 (G870S) expression (again, vs. empty) results in thousands of differentially expressed genes, only a very small % of these genes overlap with SETBP1 (G870S) enrichment sites; the authors, however, claim that the mutant is regulating gene expression through enhancement in its promoter binding to target genes. This seems to be selective reasoning, and the authors should spend more time dissecting exactly what SETBP1 (G870S) might be doing throughout the remainder of the genome. Also, In Figure 2f, the text is almost unreadable...more emphasis on Figure preparation should be placed.

It is true that we specifically and selectively focused on the effect of SETBP1 at promoter level and this is now more clearly stated in the manuscript (Discussion, first and last paragraph). However this approach allows to directly assess the ability of a DNA binding protein to act as a transcriptional modulator. In this sense, the evidence that only a minor fraction of the DEGs represents the direct target of the GOI is also not unexpected, as the overall transcriptional effect is the composition of direct (SETBP1) and indirect (e.g. transcription factors downstream of SETBP1) effects. It is also important to consider that the identification of DEGs under the direct control (promoter) of SETBP1 gave very clear results, with more than 90% of target genes showing a significant upregulation. In the opinion of the authors it is extremely unlikely that this occurred just by chance. Moreover, the results of the AT-hooks/Delta-HBM knock-out experiments indicate that, whenever we suppress the ability of SETBP1 to bind the genomic DNA or to recruit the activator complex, the upregulation of target genes is abolished; it is also worth noticing that this occurs despite the presence of the G870S mutation. The quality of the figures was improved in this revised version. We regret for this inaccuracy.

5) In Figure 3, it is very unclear as to how the authors statistically performed correlations between differential SETBP1 (G870S) enrichment vs. histone PTMs. Were Spearman rank correlations performed? If so, then the stats need to be provided.

This point is now discussed in detail in the methods section (Chromatin Immunoprecipitation sequencing section)

6) The biochemical experiments in Figure 4, as presented, are inconclusive. Again, SETBP1 (G870S) was only compared to empty vector, and not WT SETBP1. Furthermore, the inputs for all IPs are not normalized, so it is impossible for the authors to claim relative enrichment.

7) For qChIP studies, it is inappropriate to display the data as "Relative Enrichment (A.U.)." These data need to be shown as % input.

6 and 7) We agree with the reviewer: qChIP data are now shown as % input, as requested. This is now clearly stated in the y-axis of the ChIP panels. For the rationale of the comparison between mutated SETBP1 and empty vector please refer to point #1.

8) In the absence of biophysical assessments of binding (e.g., ITC assays examining recombinant SETBP1 binding to HCF1, etc.), such claims of interactions are overstated.

At the current level of analysis we were only interested in the identification of the proteins interacting with SETBP1 in a regulatory complex not in the stoichiometry of these interactions or in the calculation of the association/dissociation constant. Although we do not exclude to further investigate these interactions at very fine level, as proposed by the reviewer, at present we only focused on Co-IP and FRET assays, as these tools are commonly used to identify interactions at protein-protein level.

9) The neuronal experiments are not interpretable in the absence of WT overexpression.

10) Figures 5 and 6 should probably be presented in reverse order (i.e., cell culture first, then in vivo analyses). Having said this, including data from both seems incohesive, and the neuronal data either need to be more fully developed or removed entirely.

9 and 10) To further clarify the specific effect of SETBP1 mutations during neuron development, we generated in utero experiments for SETBP1-WT as well as SETBP1-ATH1,2 and SETBP1-DeltaHBM. These new models allowed us to confirm our quantitative hypothesis: knocking-out either the AT-Hook or the HCF1 binding site almost completely restored the correct differentiation of the cortex neurons, despite the presence of SETBP1 mutation. This highlights the role of the abnormal transcription in the onset of the SGS syndrome. A similar analysis performed using wt SETBP1 confirmed the quantitative effect of SETBP1 mutations, with neuron progenitors overexpressing wt SETBP1 showing a phenotype similar, albeit decreased, to that of mutated SETBP1. We agree that presenting Figure 5 and 6 in reverse order would improve manuscript readability (done).

In sum, although this paper has a reasonable hypothesis and provides some potentially interesting data, major revisions are needed to warrant publication at this time

Reviewers' comments:

Reviewer #1 (Remarks to the Author):

In their revised manuscript, authors address most of the concerns raised by this reviewer, except for the issue of wt-control, which should be addressed before the manuscript is considered for publication. This reviewer believes that wt SETBP1 should be included as a control, as every study addressing the role of mutant vs. wt protein, even though the authors hypothesize that the major oncogenic action of mutant SETBP1 should be explained by quantitative difference between wt and mutant proteins. In addition to show that the experimental system properly works, this is also important just to confirm that the results for mutant protein are more prominent as expected from the hypothesis. It is not sufficient to evaluate the SETBP1 ChIP-seq peak.

Reviewer #2 (Remarks to the Author):

The paper has now been much improved and can be published in Nat Communications.

Reviewer #3 (Remarks to the Author):

Overall, I feel that the authors have done a reasonably nice job in responding to most of my previous critiques, and I continue to find this study of potential interest to the field. However, I remain deeply troubled by their use of the so-called "quantitative model" to explain why a WT SETBP1 control should not be used in comparison to the mutant protein (vs. an empty vector alone). For example, in Fig. 2d, they clearly demonstrate that specific genes that are shown to be differentially regulated by G870S vs. Empty in their RNA-seq analyses are also regulated (perhaps to a lesser extent) by WT SETBP1 (e.g., CEP44), and other genes (e.g., CDKN1B) are similarly regulated by both WT and G870S vs. the empty vector control. These findings then make me suspicious of the "epigenetic modulation" data presented in Fig. 3, which also do not use the WT control. Simply stating that you don't want to use the WT control because it will lead to less differentially regulated genes ("underestimation") is not a good enough rationale. For example, if the goal is really to assess the impact of the mutation, then even if very few gene expression/epigenetic modulation differences are observed between WT vs. G870S, we would at least be confident that those genes are specifically affected by the mutation in question. Without this, the significance of the findings are challenging to interpret, especially given the fact that WT vs. mutant constructs appear to result in very similar deficits in neurodevelopment (Fig. 6) suggesting that SETBP1 overexpression itself has much more of an effect than the mutation. In sum, I continue to believe that the addition of the WT control to the various analyses where it is currently excluded will be necessary prior to being suitable for publication.

Response to Reviewers' comments:

Reviewer #1 (Remarks to the Author):

In their revised manuscript, authors address most of the concerns raised by this reviewer, except for the issue of wt-control, which should be addressed before the manuscript is considered for publication. This reviewer believes that wt SETBP1 should be included as a control, as every study addressing the role of mutant vs. wt protein, even though the authors hypothesize that the major oncogenic action of mutant SETBP1 should be explained by quantitative difference between wt and mutant proteins. In addition to show that the experimental system properly works, this is also important just to confirm that the results for mutant protein are more prominent as expected from the hypothesis. It is not sufficient to evaluate the SETBP1 ChIP-seq peak.

We thank the reviewer for this helpful criticism. Although in this specific context we consider of critical importance to highlight the difference between a standard model and a quantitative one to avoid confusion, we also agree on the importance of adding the wt as a control. To summarize, in this revised version the wt control was added for the following experiments:

- ChIP-Seq (new)
- RNA-Seq (new)
- ATAC-Seq
- Mass-Spectrometry/HPLC
- FRET
- CHIP (for a subset of DNA targets, new)
- Q-PCR (new)
- Epigenetic marks (H3K9Ac, new)
- MECOM analysis (new)
- Mouse embryo experiments

Figures and text were modified accordingly. In all cases and with no exception we were able to confirm the quantitative model, which, we think, gives further value to this work. We sincerely hope that the new experiments will address the remaining concerns of the reviewer.

Reviewer #2 (Remarks to the Author):

The paper has now been much improved and can be published in Nat Communications.

We thank the reviewer for this positive evaluation.

Reviewer #3 (Remarks to the Author):

Overall, I feel that the authors have done a reasonably nice job in responding to most of my previous critiques, and I continue to find this study of potential interest to the field. However, I remain deeply troubled by their use of the so-called "quantitative model" to explain why a WT SETBP1 control should not be used in comparison to the mutant protein (vs. an empty vector alone). For example, in Fig. 2d, they clearly demonstrate that specific genes that are shown to be differentially regulated by G870S vs. Empty in their RNA-seq analyses are also regulated (perhaps to a lesser extent) by WT SETBP1 (e.g., CEP44), and other genes (e.g., CDKN1B) are similarly regulated by both WT and G870S vs. the empty vector control. These findings then make me suspicious of the "epigenetic modulation" data presented in Fig. 3, which also do not use the WT control. Simply stating that you don't want to use the WT control because it will lead to less differentially regulated genes ("underestimation") is not a good enough rationale. For example, if the goal is really to assess the impact of the mutation, then even if very few gene expression/epigenetic modulation differences are observed between WT vs. G870S, we would at least be confident that those genes are specifically affected by the mutation in question. Without this, the significance of the findings are challenging to interpret, especially given the fact that WT vs. mutant constructs appear to result in very similar deficits in neurodevelopment (Fig. 6) suggesting that SETBP1 overexpression itself has much more of an effect than the mutation. In sum, I continue to believe that the addition of the WT control to the various analyses where it is currently excluded will be necessary prior to being suitable for publication.

We thank the reviewer for this helpful criticism. In this revised version we made a significant effort in order to improve the manuscript in the direction indicated by the reviewer. Therefore, wt controls were added for virtually all the main experiments. In summary, wt controls are now provided for CHIP-Seq, RNA-Seq, ATAC-Seq, Mass-Spectrometry/HPLC, FRET, CHIP (for a subset of DNA targets identified by CHIP-Seq), Q-PCR, H3K9Ac epigenetic mark, MECOM analysis, Mouse embryo experiments. However we would like to point out that our reluctance in adding a 'wt control' in this context doesn't steam from laziness or because we want to hide unexpected results. Instead, the reason is that in the experimental models we used, the expression of SETBP1, both wt and G870S, is significantly increased when compared to naive cells (e.g. Figure S1). Unfortunately, at the time being, this bias is basically unavoidable due to the non-physiological promoter used. The low level of expression of SETBP1 and the lack of good (or even decent) quality anti-SETBP1 antibodies prevents the use of other techniques, such as CRISPR/Cas9, to generate and use more physiological models. We acknowledge this limitation in the results section, 'SETBP1 as a DNA binding protein' paragraph.

Taking into account the quantitative effect of SETBP1 mutations (i.e. they stabilize the protein), even the term 'wt' could be misleading and may generate confusion in readers that are not deeply accustomed to SETBP1 biology and to the peculiar effect of SETBP1 somatic mutations.

Hence, your statement "SETBP1 overexpression itself has much more of an effect than the mutation" is absolutely true. Rephrasing it, we could say that SETBP1 mutations represent a 'phenocopy' of SETBP1 overexpression. To further confirm this point, we generated a small bioinformatics tool which allows us to test the linear correlation of different ChIP-Seq BigWig tracks (details can be found in methods) and we applied it to G870S and WT anti-V5 ChIP-Seq data and to G870S anti-H3K9Ac and a newly generated WT anti-H3K9Ac. In both cases the linearity was excellent ($r=0.985$ and $p<0.00001$ for anti-V5; $r=0.873$ and $p<0.00001$ for anti-H3K9Ac), which, again, confirms the quantitative 'phenocopy' model. These data are now reported in supplementary figure 4 and described in the results section, paragraph: 'SETBP1 is part of a multiprotein epigenetic activator complex'.

To this end, while in this revision we made significant efforts in adding wt controls, at the same time we also tried to clarify this point in the text by highlighting the peculiar meaning of these experiments.

The fact that clinical (aCML patients) samples confirm the results obtained in less physiological conditions strengthens, in our opinion, the general validity of the results presented.

Moreover, the "quantitative model" hypothesis is also supported by a recently published paper (Acuna-Hidalgo R. et al, PLoS Genet. 2017 Mar 27;13(3):e1006683. doi: 10.1371/journal.pgen.1006683).

In conclusion, while the effect of SETBP1 overexpression (either WT or mutated) on the level of transcription of a single gene could be due to the model utilized, the overall effects, the uniform pattern observed (increased transcription), the mechanistic data provided (identification of a multiprotein complex), and the confirmation of these results using clinical samples where SETBP1 is under its physiological promoter, provide in our opinion sufficient evidence of the general validity of the model proposed in our paper.

REVIEWERS' COMMENTS:

Reviewer #1 (Remarks to the Author):

In this revision, to answer this reviewer's concerns, authors included wt control in many experiments where it is necessary. This reviewer has no further concerns, except for some minor points as described below.

- 1) However, the results of the analyses including wt control are not fully described for some experiments. For example, in Fig. 5c and Fig. 6, no comparison was made between mutant and wt.
- 2) If the difference between wt and mutant is only quantitative, why the enriched motifs were different between the two in Fig. 1b?
- 3) Why no wt controls were included in the experiments presented in Fig. 4.
- 4) In Fig. 6d, the behaviors of wt (black) and mutant (red) were different between different bins: in bins 1 and 5, the percentage of wild-type was larger than mutant, whereas the percentages were smaller for wild-type, compared to the mutant. If the difference is quantitative, one expects similar trend between wild-type and mutant.

Reviewer #3 (Remarks to the Author):

The authors' inclusion of wt controls, as well as further clarifications regarding the model being used, is greatly appreciated. This manuscript is thus greatly improved over previous versions and is now suitable for publication in Nature Communications.

Response to Reviewers' comments:

Reviewer #1 (Remarks to the Author):

In this revision, to answer this reviewer's concerns, authors included wt control in many experiments where it is necessary. This reviewer has no further concerns, except for some minor points as described below.

1) However, the results of the analyses including wt control are not fully described for some experiments. For example, in Fig. 5c and Fig. 6, no comparison was made between mutant and wt.

3) Why no wt controls were included in the experiments presented in Fig. 4.

As the time required to include a truly complete list of wt controls would be very large and timing is also a critical factor in order to publish scientific works, we opted for a compromise. We generated WT controls for a very large number of experiments and notably for all the critical ones. Specifically:

- **For the experiment in fig. 5c the WT control was added for the CHIP-Seq experiment and the results can be seen in the main figure (again, they confirm the quantitative model). We didn't reply the confirmative CHIP followed by Q-PCR (the one in the box in fig. 5c) as the results for G870S and Empty were perfectly in line with the CHIP-Seq experiments.**
- **For figure 6: wt data were generated by performing new in utero experiments for the WT as well as for the AT-hooks knock-out and for the Delta-HBM: results were compared in figure 6 panel d.**
- **We didn't add wt controls for figure 4; however mass spectrometry data were generated for wt together with G870S and Empty samples. In this specific case we decided to focus on the G870S vs empty as we considered the experiments in figure 4 mostly confirmatory.**

2) If the difference between wt and mutant is only quantitative, why the enriched motifs were different between the two in Fig. 1b?

In a quantitative model we expect that differences do exist between wt and mutant, given that the suppression of the proteasomal degradation of the mutant SETBP1 protein leads to an increase in protein amount. However, the significant overexpression of the SETBP1 cDNA mediated by the strong promoter that is present in our constructs causes the wt to be closer to the mutant than it would be in a more physiological model. Specifically, in the enriched motif plots shown in figure 1b the consensus motif generated for wt and mutant SETBP1 is virtually identical: the whole 'AAAAT' motif is shared between wt and mutant; the only differences between the two are in the final 'AT' sequence, where the presence of a final Thymidine in the mutant corresponds to a Thymidine/Adenine in the wt and the A/G in the mutant corresponds to an A in the wt in penultimate position.

4) In Fig. 6d, the behaviors of wt (black) and mutant (red) were different between different bins: in bins1 and 5, the percentage of wild-type was larger than mutant, whereas the percentages were smaller for wild-type, compared to the mutant. If the difference is quantitative, one expects similar trend between wild-type and mutant.

In the opinion of the authors a similar trend do exist for wt and mutant SETBP1 in the experiments shown in figure 6d: if the 'cortex bins' (e.g. 5 and 4) are taken together, a critical and well-evident pattern can be seen, where the presence of an active and overexpressed SETBP1 (i.e. mutant or wt) greatly impairs the ability of neuron progenitors to efficiently migrate/differentiate into cortical neurons. In cases where SETBP1 is overexpressed in presence of AT-Hooks mutations or in presence of a deletion of the HCF1 binding site, the overexpression of SETBP1 is no longer able to impair this mechanism and therefore neurons are found in bins 4 and 5 up to a level that is comparable to the GFP control (green).

Reviewer #3 (Remarks to the Author):

The authors' inclusion of wt controls, as well as further clarifications regarding the model being used, is greatly appreciated. This manuscript is thus greatly improved over previous versions and is now suitable for publication in Nature Communications.

We thank the reviewer for this positive evaluation.